# Endocytic sorting motif interactions involved in Nef-mediated downmodulation of CD4 and CD3

Santiago Manrique[1,2], Daniel Sauter[3], Florian A. Horenkamp[2], Sebastian Lülf[2,4], Hangxing Yu[3], Dominik Hotter[3], Kanchan Anand[1,4], Frank Kirchhoff[3] & Matthias Geyer[1,2,4]

Lentiviral Nefs recruit assembly polypeptide complexes and target sorting motifs in cellular receptors to induce their internalization. While Nef-mediated CD4 downmodulation is conserved, the ability to internalize CD3 was lost in HIV-1 and its precursors. Although both functions play key roles in lentiviral replication and pathogenicity, the underlying structural requirements are poorly defined. Here, we determine the structure of $SIV_{mac239}$ Nef bound to the ExxxLM motif of another Nef molecule at 2.5 Å resolution. This provides a basis for a structural model, where a hydrophobic crevice in simian immunodeficiency virus (SIV) Nef targets a dileucine motif in CD4 and a tyrosine-based motif in CD3. Introducing key residues into this crevice of HIV-1 Nef enables CD3 binding but an additional N-terminal tyrosine motif is required for internalization. Our resolution of the CD4/Nef/AP2 complex and generation of HIV-1 Nefs capable of CD3 downregulation provide insights into sorting motif interactions and target discrimination of Nef.

[1] Institute of Innate Immunity, Department of Structural Immunology, University of Bonn, Sigmund-Freud-Str. 25, 53127 Bonn, Germany. [2] Max Planck-Institute of Molecular Physiology, Department Physical Biochemistry, Otto-Hahn-Str. 11, 44227 Dortmund, Germany. [3] Institute of Molecular Virology, Ulm University Medical Center, Meyerhofstr. 1, 89081 Ulm, Germany. [4] Center of Advanced European Studies and Research (caesar), Ludwig-Erhard-Allee 2, 53175 Bonn, Germany. Correspondence and requests for materials should be addressed to M.G. (email: matthias.geyer@uni-bonn.de)

The Nef protein of human (HIV) and simian (SIV) immunodeficiency viruses is an important replication and pathogenicity factor. Individuals infected with *nef*-deficient HIV-1 strains show very low viral loads and become long-term survivors without antiretroviral therapy[1, 2]. Nef increases virion infectivity by counteracting the cellular restriction factor SERINC5[3, 4] and facilitates viral immune evasion by decreasing the cell surface levels of several immune receptors, such as CD3, CD4, and MHC class I[5–7]. Nef-mediated internalization of CD4 from the surface of infected cells is highly conserved among primate lentiviruses and thought to promote the release of fully infectious viral particles and to prevent superinfection[8–10]. The Nef proteins of HIV-2 and most SIVs also efficiently internalize CD3, an essential cofactor in T cell receptor (TCR) signaling. This specific Nef function was entirely lost by HIV-1 and its simian precursors[11]. Nef-mediated internalization of CD3 allows HIV-2 and most SIVs to disrupt formation of the immune synapse between infected CD4+ T cells and primary antigen-presenting cells[12]. HIV-1 Nef proteins also dysregulate the immune synapse[13, 14] and modulate downstream TCR signaling pathways[15]. However, SIV and HIV-2 Nef proteins that downmodulate CD3 disrupt immune synapse function and TCR-dependent T cell activation much more severely than HIV-1 Nefs[12] and may thus suppress the hyperactivation of the immune system that drives progression to AIDS in HIV-1 infected individuals and is absent in natural SIV infection[16]. It has been shown that CD3 downmodulation depends on residues within the conserved core region of Nef[17, 18]. However, the structural basis for the fundamental differences in the ability of primate lentiviral Nef proteins to endocytose CD3 remained unknown.

CD4 and the TCR–CD3 complex are constantly internalized via clathrin-mediated endocytosis, also in the absence of infection. This involves the interaction between sorting motifs in the cytoplasmic tails of these receptors and the assembly polypeptide 2 (AP2) complex[19, 20]. AP2 binds to clathrin and activates triskelion formation of clathrin-coated vesicles at the inner cell membrane. The heterotetrameric AP2 complex is composed of subunits α, β2, μ2 and σ2[21]. It harbors a recognition site for the tyrosine-based sorting motif Yxxϕ (where x is any amino acid and ϕ either L, I, V, M or F) in the μ2 subunit and a recognition site for the dileucine-based sorting motif (E/D)xxxLϕ in the σ2/α hemicomplex. CD4 contains a non-canonical dileucine-based sorting motif (SQIKRLL) in its C-terminal cytoplasmic tail. The serine residue upstream of the LL motif needs to be phosphorylated in order to mimic the negatively charged glutamate or aspartate residues of a classical dileucine motif and to be recognized by AP2[22]. Interestingly, Nef is able to adapt CD4 to clathrin-mediated endocytosis in the absence of serine phosphorylation[23]. In contrast to CD4, the ζ chain of the CD3 complex contains multiple internalization motifs in its cytoplasmic tail including three copies of an immunoreceptor tyrosine-based activation motif (ITAM) with the consensus Yxx(L/I)x$_{6–9}$Yxx(L/I). The tyrosines of the ITAMs are phosphorylated upon TCR ligation by Src family kinases and mediate downstream signal transduction and T cell activation[24]. Lentiviral Nefs that downmodulate CD3 directly interact with the cytoplasmic tail of the ζ chain. The interaction sites overlap with the ITAM regions and were mapped to the sequence motifs Y$_{72}$NELNL and Y$_{123}$SEIGM, termed SIV Nef interaction domain 1 and 2 (or SNID1 and SNID2)[25].

Nef stimulates clathrin-mediated endocytosis of CD3, CD4 and other cellular receptors by acting as an adapter of the adapter protein machinery[26]. It contains an N-terminal membrane anchor domain of 60–120 amino acids length followed by a core domain of 130–150 amino acids, containing a highly conserved dileucine-based sorting motif at the center of a C-terminal flexible

**Table 1 Data collection and refinement statistics**

| | SIV$_{mac239}$ Nef–Hck$_{SH3-E}$ | SIV$_{mac239}$ Nef |
|---|---|---|
| *Data collection* | | |
| X-ray source | Synchrotron | Synchrotron |
| Beam line | SLS X06SA | SLS X10SA |
| Wavelength (Å) | 1.0000 | 0.9778 |
| Temperature (K) | 100 | 100 |
| *Crystal information* | | |
| Resolution (Å)$^a$ | 2.78 | 2.50 |
| Space group | $P3_2$ | $C222_1$ |
| Unit cell dimensions | | |
| a, b, c (Å) | a = 104.0, b = 104.0, c = 53.0 | a = 51.50, b = 140.70, c = 106.40 |
| α, β, γ (°) | α = β = 90, γ = 120 | α = β = γ = 90 |
| $R_{means}$ | 0.148 (0.96) | 0.060 (1.0) |
| Mean I/σ$_I$ | 10.7 (2.0) | 20.0 (2.1) |
| Completeness (%) | 100 | 99.9 |
| *Refinement* | | |
| Unique reflections | 16,118 | 13,775 |
| No. of reflections used | 16,114 | 12,248 |
| Model contents | A: SIV$_{mac239}$ Nef (103–233, Δ183–187) B: SIV$_{mac239}$ Nef (103–233) C: SH3$_E$ (81–134) D: SH3$_E$ (80–134) | A: SIV$_{mac239}$ Nef (107–233, Δ184–202) B: SIV$_{mac239}$ Nef (104–233) |
| $R_{work}$/$R_{free}$ | 0.191/0.228 | 0.205/0.257 |
| *Number of atoms* | | |
| Protein | 3009 | 1962 |
| Water | 55 | 49 |
| *B factors* | | |
| Protein (avg. B value) (Å$^2$) | 52.62 | 73.2 |
| Water (avg. B value) (Å$^2$) | 48.53 | 72.2 |
| *R.m.s deviations* | | |
| R.m.s. deviations bonds (Å) | 0.009 | 0.009 |
| R.m.s. deviations angles (°) | 1.027 | 1.40 |
| Ramachandran plot (%) | Most favored: 98.88 | Most favored: 96.28 |
| | Allowed: 1.12 | Allowed: 3.72 |
| PDB accession code | 5NUH | 5NUI |

$^a$Values in parentheses correspond to the highest resolution shell (2.55–2.50 Å or 2.83–2.78 Å, respectively)

loop[27]. This exposed sorting motif enables Nef to interact with AP2 and to induce the internalization of its target receptors. Although Nef structures have been determined by crystallography and nuclear magnetic resonance (NMR) spectroscopy[28–30], the molecular basis of Nef binding to a dileucine-based sorting motif such as that of CD4 is still missing, potentially due to the low affinity of this endocytic interaction. Furthermore, the structural basis for the fundamental differences in the ability of primate lentiviral Nefs to downmodulate CD3 is poorly understood.

Here, we examined by structural means how Nef interacts with dileucine-based and tyrosine-based sorting motifs. We solved the structure of SIV$_{mac239}$ Nef bound to the ExxxLM motif of another Nef molecule and took advantage of known Nef structures to reconstitute the endocytic CD4/Nef/AP2 complex as it assembles at the plasma membrane. The architecture of the trimeric complex reveals that the dileucine motif recognition site of Nef

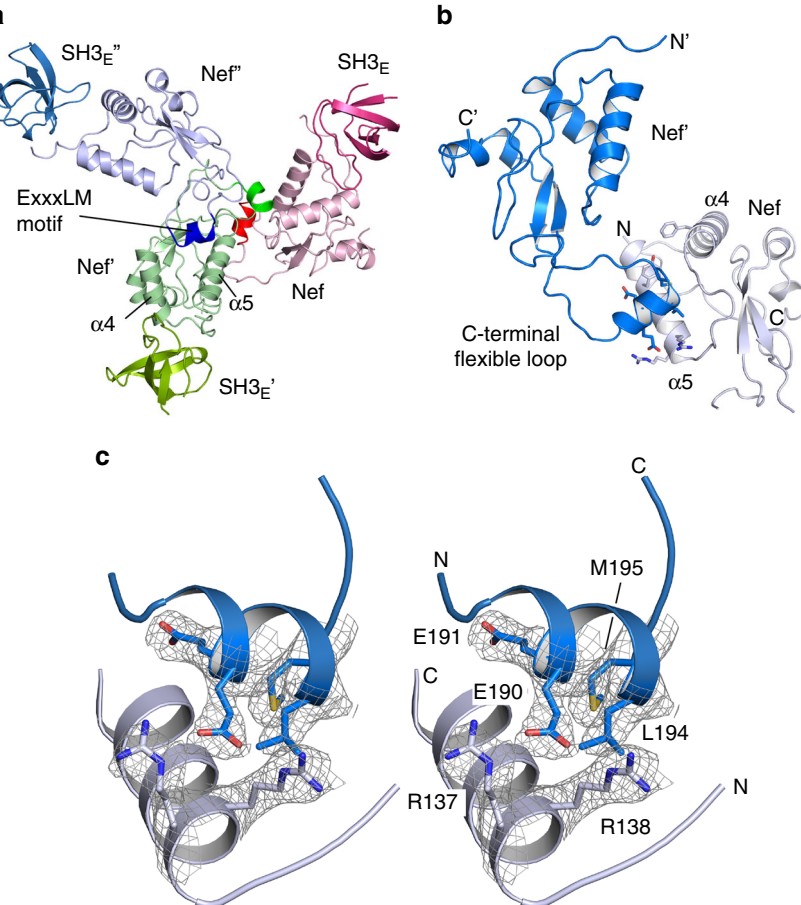

**Fig. 1** Assembly of SIV$_{mac239}$ Nef by intermolecular sorting motif interactions. **a** Structure of the Nef-SH3 heterodimer assembly in a hexagonal lattice. The dileucine-based sorting motif (colored *bold*) in the C-terminal flexible loop of Nef contacts the hydrophobic sorting motif recognition site (colored *light*) of a neighboring Nef/SH3 heterodimer in a three-fold symmetry. **b** Structure of SIV$_{mac239}$ Nef at 2.5 Å resolution. The E$_{190}$EHYLM sorting motif of one Nef subunit interacts with the sorting motif recognition site of another Nef subunit. The helical conformation of the sorting motif is shown as ribbon diagram (*blue*) with residues EExxLM displayed as sticks. **c** Stereo image of the final 2F$_o$–F$_c$ electron density map of the dileucine-based sorting motif displayed at 1σ. Residues E$_{191}$ExxLM of the sorting motif and the two arginines R137 and R138 of the recognition site are shown

is closer to the plasma membrane than the dileucine motif recognition site of AP2, explaining how Nef acts as an adapter in a cascade of interactions between CD4–Nef and Nef–AP2 complexes. We also reconstitute the long-lost CD3 downregulation function of HIV-1 Nefs by introducing a tyrosine-based sorting motif in the N-terminal domain and exchanging residues in the hydrophobic crevice of the core domain. Our study thus provides insights into the molecular basis for the versatility of Nef as an adapter protein for the internalization of surface molecules containing endocytic sorting motifs.

## Results

**Crystal structures of SIV$_{mac239}$ Nef.** To gain insights into the binding of Nef to the dileucine-based sorting motif of CD4, we attempted to co-crystallize HIV-1$_{SF2}$ or SIV$_{mac239}$ Nef proteins with peptides of the CD4 cytoplasmic domain with and without an SH3 domain as stabilizing factor. The complex between SIV$_{mac239}$ Nef 66–235 (SIV-B) and the SH3 domain of human Hck 80–140 (SH3$_E$), engineered in the RT loop for improved Nef binding[31], yielded diffracting crystals. The structure was determined by molecular replacement at 2.78 Å resolution with good stereochemistry (Table 1). The model contains residues 103–233 of SIV Nef and residues 80–134 of the SH3 domain, whereas no electron density was identified for

the CD4 peptide (Fig. 1a). The asymmetric unit consists of two Nef/Hck heterodimers that assemble at the Nef–SH3 domain interface with the C-terminal flexible loop of Nef exposed in opposite directions (Supplementary Fig. 1). At the contact interfaces a three-fold rotational symmetry appears with each C-terminal flexible loop of Nef contacting the sorting motif recognition site of the neighboring Nef–SH3 heterodimer (Fig. 1a). In this crystallographic assembly, the E**EHYLM** sorting motif of the C-terminal flexible loop interacts with the hydrophobic crevice formed by the two central helices α4 and α5 of Nef and the interconnecting loop. Intriguingly, this interaction displays the structural basis for the binding of a dileucine sorting motif to Nef.

We were able to improve the crystal structure to 2.50 Å resolution by using an N-terminally truncated SIV$_{mac239}$ Nef starting at M87 before the extended acidic cluster in SIV Nef (Supplementary Fig. 2). As before, a synthetic peptide of 16 residues comprising the cytoplasmic dileucine motif of CD4 was added to the crystallization screens. Again, the sorting motif of one Nef molecule in the asymmetric units contacts the hydrophobic crevice of another Nef molecule (Fig. 1b). In this structure, the electron density map was visible for all residues in the dileucine-based sorting motif, providing detailed molecular insights into the interaction of Nef with endocytic sorting motifs (Fig. 1c). Additional density of a helical conformation is visible in

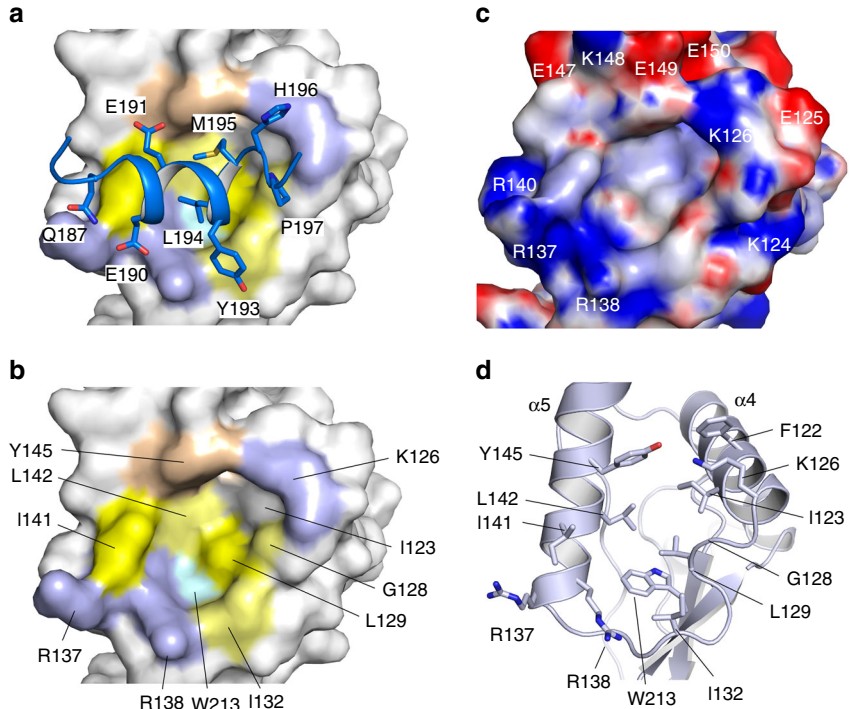

**Fig. 2** Binding of the dileucine-based sorting motif to Nef. **a** Binding of the ExxxLM sorting motif (*blue*) to the hydrophobic crevice of Nef shown as surface representation. Residues of the $Q_{187}$ED**EE**HY**LM**H sequence that mediate direct interactions with Nef are shown as *sticks* and labeled. **b** Mapping of the interaction surface in Nef. Hydrophobic residues I123, L129, W213, the methylene side chain groups of R138, and L142 form the core of the binding pocket. Residues K126, G128, I132, R138, R137, I141 and Y145 delineate the surrounding of the hydrophobic crevice. **c** Electrostatic surface potential of SIV$_{mac239}$ Nef displayed from −8 $k_B T$ (*red*) to +8 $k_B T$ (*blue*). The surrounding of the hydrophobic crevice is largely positively charged by residues K124, K126, R137, R138 and R140. **d** The hydrophobic crevice of Nef. Residues that interact with the ExxxLM sorting motif are shown in *stick* representation

a solvent channel between both Nef subunits. This sequence could not be unambiguously assigned as only the main chain of two helical turns is defined. Overall, the structures of the core domains of HIV-1$_{SF2}$ and SIV$_{mac239}$ Nef are similar with an RMS deviation of 0.84 Å for an overlay of 91 C$^\alpha$ atoms, reflecting 51% sequence identity in this region. Notably, the Nef protein of SIVmac is structurally highly similar to HIV-2 Nefs because both viruses belong to the same lineage of primate lentiviruses and share a common origin[32]. HIV-2 is the result of at least nine independent zoonotic transmissions of SIVs infecting sooty mangabeys and some HIV-2 strains are genetically more closely related to SIVmac than to other HIV-2 isolates. As observed for HIV-1 Nef and its myristoylated full length variant[33, 34], SIV$_{mac239}$ Nef eluted as monomer in size exclusion chromatography experiments or as heterodimer in complex with the SH3 domain (Supplementary Fig. 3). Thus, the interaction of the ExxxLM sorting motif of one Nef molecule with the sorting motif recognition domain of another Nef molecule does not lead to multimerization in vitro.

**Molecular interactions of the dileucine motif with Nef.** Upon binding to the hydrophobic crevice, the ExxxLM sorting motif of SIV$_{mac239}$ Nef forms a regular α-helix (αL) of two turns ranging from E188 to M195, thereby ending with the last canonical residue of the internalization motif. Residues E190 and L194 are exposed on the same side of the helix and adopt key positions in the interaction with the sorting motif recognition site of Nef (Fig. 2a). The negatively charged hydroxyl side chain group of E190 aligns between the two positively charged guanidinium ion groups of arginines 137 and 138, forming two strong salt bridges in the assembly between the sorting motif and its recognition domain. L194 binds central to the hydrophobic crevice with contacts

formed to G128, L129, I132 and R138 of Nef. At a distance of 3.8 Å, L194 interacts with the Cζ3 atom of W213, which delineates the base of the hydrophobic crevice (Fig. 2b). The second hydrophobic residue of the sorting motif, M195, interacts with L142, Y145 and K126 of Nef, completing the interaction scheme of the ExxxLφ motif with its recognition domain.

Besides the three canonical residues E$_{190}$xxxL$_{194}$M$_{195}$ of the endocytic sorting motif, additional contacts to the recognition domain are formed by residues Q187, E191, Y193, H196, and P197. Whereas L194 and M195 deeply interact with the hydrophobic crevice of Nef, all other residues form contacts at the surrounding surface which is mostly positively charged (Fig. 2c). Within a distance of 3.7 Å, nine residues of Nef are directly involved in the interaction with the ExxxLφ motif, while W213 and I123 are slightly more distant. In a clockwise listing I123, L129, W213 and L142 constitute the pocket of the hydrophobic crevice, whereas K126, G128, I132, R138, R137, I141 and Y145 form the rim of the sorting motif recognition site (Fig. 2d). Of note, no water molecules were seen in the binding interface between Nef and the sorting motif, indicating that the side chains of L194 and M195 sufficiently fill the hydrophobic crevice.

**Comparison of Nef binding to CD3 ζ and CD4.** The interaction of the Nef core domain with the ExxxLM sorting motif in the C-terminal loop of another Nef molecule allows us to envision binding of Nef to the dileucine-based sorting motif of CD4. Whereas the EG E$_{164}$NNS**LL**HP sorting motif in HIV-1 Nef$_{SF2}$ and the ED E$_{190}$EHY**LM**HP sorting motif in SIV$_{mac239}$ Nef are constitutively active due to the presence of adjacent acidic patches (see the alignment in Supplementary Fig. 2), the MS**Q**$_{434}$IKR**LL**SE sorting motif in the cytoplasmic tail of CD4 needs to be phosphorylated at S433 to enable the interaction with AP2[22]. In

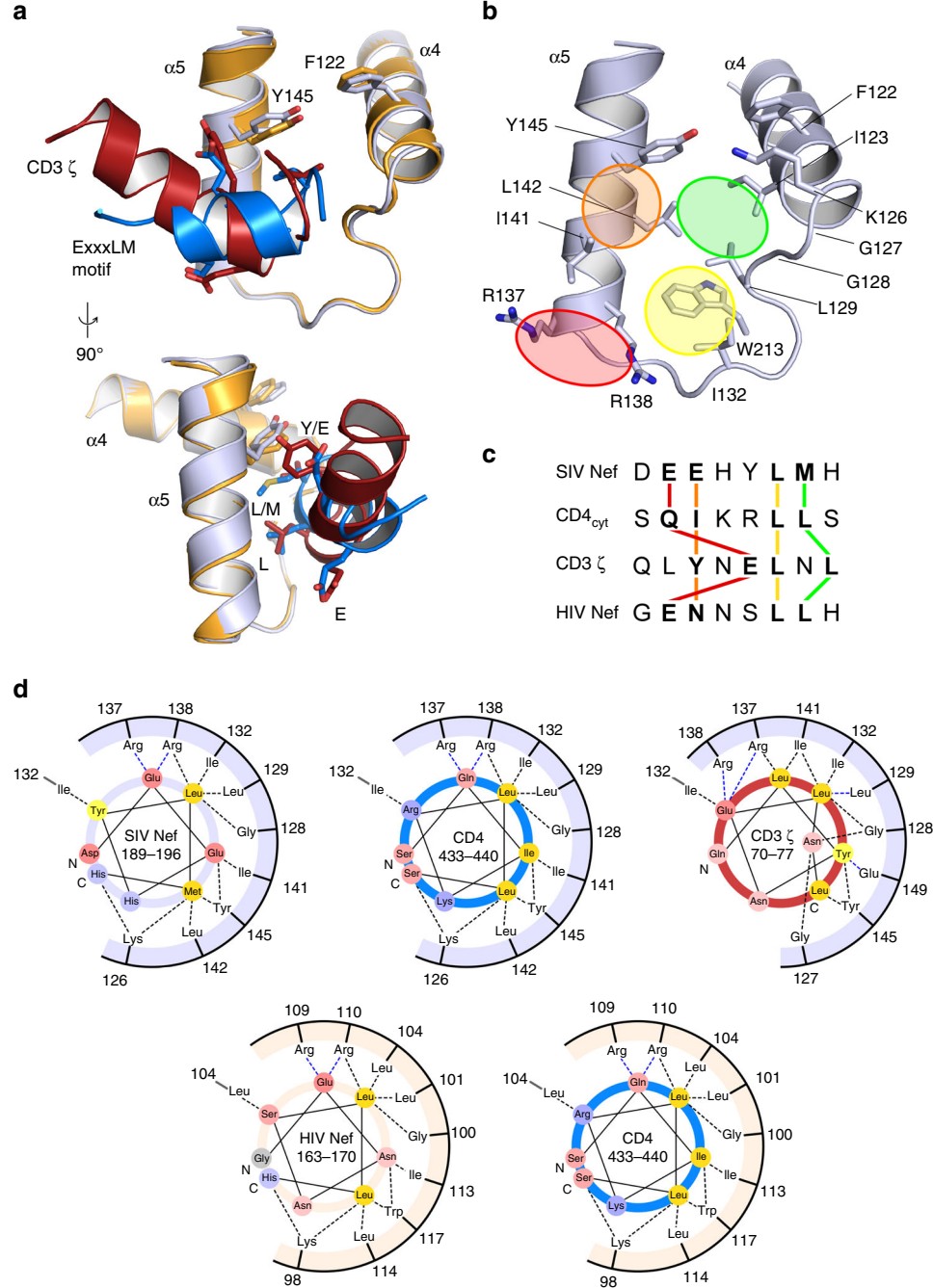

**Fig. 3** Recognition of CD3 and CD4 sorting motifs by SIV Nef. **a** Overlay of SIV$_{mac239}$ Nef crystal structures binding to the YxELxL sorting motif as contained in the CD3 ζ chain (3IK5)[30] or the ExxxLφ motif as present in CD4. Both sorting motifs are arranged on helices whose orientations vary by 48°. The key interacting residues YxELxL and EExxLM of both helices are shown as *sticks*. **b** Display of the key interaction sites in Nef. The first leucine is the determining residue that aligns central in the hydrophobic crevice to L129, I132, R138 and W213 (*yellow*). The second leucine aligns to the I123, G128, L129, L142 patch, while also bulky hydrophobic residue can be accommodated in this site (*green*). The acidic residue of the sorting motif binds to the conserved arginines (*red*), while the fourth moiety interacts with I141, L142 and Y145 (*orange*). **c** Alignment of dileucine-based sorting motifs from SIV Nef$_{mac239}$, HIV-1 Nef$_{SF2}$, and the CD4 cytoplasmic tail with the first ITAM motif of CD3 ζ. The interaction pattern with the Nef protein is indicated by *colored bars*. **d** Interaction scheme of leucine-based sorting motifs with Nefs. The endocytic sorting motif is displayed as helical wheel in the *center*, while residues of the Nef core domain mediating contacts with the sorting motifs are displayed in the *outer wheel*. Hydrogen bonds and salt bridges are drawn by *blue dashed lines*, hydrophobic interactions are colored *black*. The endocytic sorting motifs D**EE**HY**LM**H (SIV$_{mac239}$ Nef), G**EN**NS**LL**H (HIV-1 Nef$_{SF2}$), S**QI**KR**LL**S (CD4), and QL**YN**EL**N**L (CD3 ζ SNID1) are shown in the *inner helical wheels*

contrast, the interaction of CD4 with Nef is phosphorylation-independent[23]. The crystal structure of SIV$_{mac239}$ Nef with the first ITAM motif of CD3 ζ has been reported[35], providing a basis for comparing binding of CD4 and CD3 to SIV Nef. A superimposition of the SIV$_{mac239}$ Nef structures bound

to either the YxELxL sorting motif (SNID1) of CD3 ζ (PDB accession code 3IK5)[35] or bound to the ExxxLφ motif shows that both motifs adopt a helical conformation when binding to Nef (Fig. 3a). Yet, whereas the backbone structure of the Nef core domain remains identical, the orientation of the

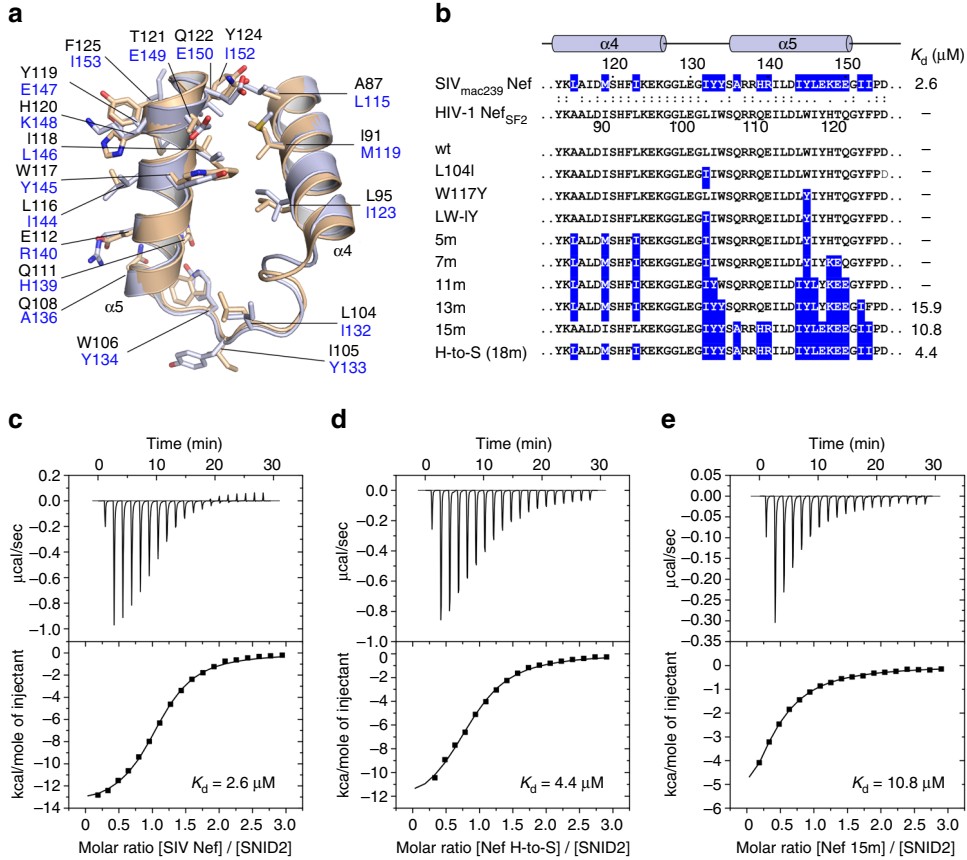

**Fig. 4** Generation of HIV-1 Nef mutants for CD3 ζ binding. **a** Overlay of crystal structures SIV$_{mac239}$ Nef (*bluewhite*, this study) and HIV-1$_{SF2}$ Nef (*wheat*, 3RBB) shows the high conformational similarity of the hydrophobic crevice in both proteins formed by helices α4 and α5. Eighteen residues vary between both proteins whose side chains are shown in *stick* representation. **b** Sequence alignment of SIV$_{mac239}$ and HIV-1 SF2 Nef proteins in the region interacting with endocytic sorting motifs. In 9 gradual steps, all 18 residues were exchanged from HIV-1 Nef to the SIV counterpart, shown in *blue*. The binding affinity to CD3 ζ determined by ITC measurements is listed on the right. **c** ITC measurement of CD3 ζ SNID2 peptide injected to SIV$_{mac239}$ Nef (66–235) showed a dissociation constant of 2.6 μM. **d** Exchange of all varying 18 residues in HIV-1 Nef to SIV led to an affinity of 4.4 μM for CD3 ζ. **e** Changing only 15 residues in Nef reduced the dissociation constant to 10.8 μM

helix relative to the hydrophobic crevice of Nef is markedly twisted by a turn of 48° relative to each other (Supplementary Fig. 4). A display from two different perspectives shows that the major leucine in both sorting motifs is positioned at exactly the same binding site in Nef (Fig. 3a, *bottom*). The same holds true for the carboxyl groups of the acidic glutamate, which align between the two conserved arginines of Nef. As this residue is one position upstream relative to the central leucine in the YxELxL motif and three positions upstream in the ExxxLM motif, both side chains point from two different angles to the same position in Nef (Fig. 3a, *top*). Likewise, the second hydrophobic residue is placed at the same position toward Nef, as CD3 ζ loops out from the helical fold with the asparagine of the YxEL**N**L sequence bridging the distance to the second leucine. Finally, the backbone Cα position of the Y in CD3 ζ is almost identical to those of the second glutamate in the E**E**xxLM motif as both residues are in the same distance to the central leucine.

The superimposition of the YxELxL and EExxLM sorting motifs allows mapping of the interaction sites in SIV$_{mac239}$ Nef (Fig. 3b). This comparison suggests a structure-based alignment of the dileucine-based sorting motifs in SIVmac Nef, HIV-1 Nef, and CD4 with the tyrosine-based ITAM sorting motif of CD3 ζ (Fig. 3c). It shows the central position of the highly conserved leucine, whereas the other signature determining residues of the sorting motifs align to the surfaces of the Nef recognition site according to the colored lines. A contact map displaying the

sorting motif as helical wheel in the center surrounded by interacting residues of the Nef core domain shows how the conserved residues of these endocytic motifs adopt similar positions in the binding interface (Fig. 3d). Based on the intermolecular interaction of the SIV Nef sorting motif with Nef, we anticipate how the CD4 sorting motif binds to SIV Nef and compare this to binding of CD3 ζ to SIV Nef (*upper panel*, left to right). We transfer these interactions to HIV-1 Nef and anticipate by sequence alignment the possible recognition of Nef to its own ExxxLL sorting motif or the SQxxxLL sorting motif of CD4 (Fig. 3d, *lower panel*). In agreement with our model, mutation of the di-arginine motif, which forms a specific salt bridge with the glutamate of the ExxxLM motif, does not only abrogate CD4 downmodulation by HIV-1 Nef[36, 37], but also resulted in a loss of the CD3 and CD4 downmodulation activities of SIVmac, SIVsmm, and HIV-2 Nef (Supplementary Fig. 5). Intriguingly, Nef mutants lacking the di-arginine motif still efficiently decrease MHC-I surface levels. Together with our structural data, these functional analyses illustrate the versatility of Nef proteins from HIV-1 and SIV as adapters for the internalization of various endocytic sorting motifs.

**Design of an HIV-1 Nef variant competent for CD3 ζ binding.** The alignment of SIV Nef binding to the YxELxL and ExxxLM sorting motifs provides a molecular basis for recognition and

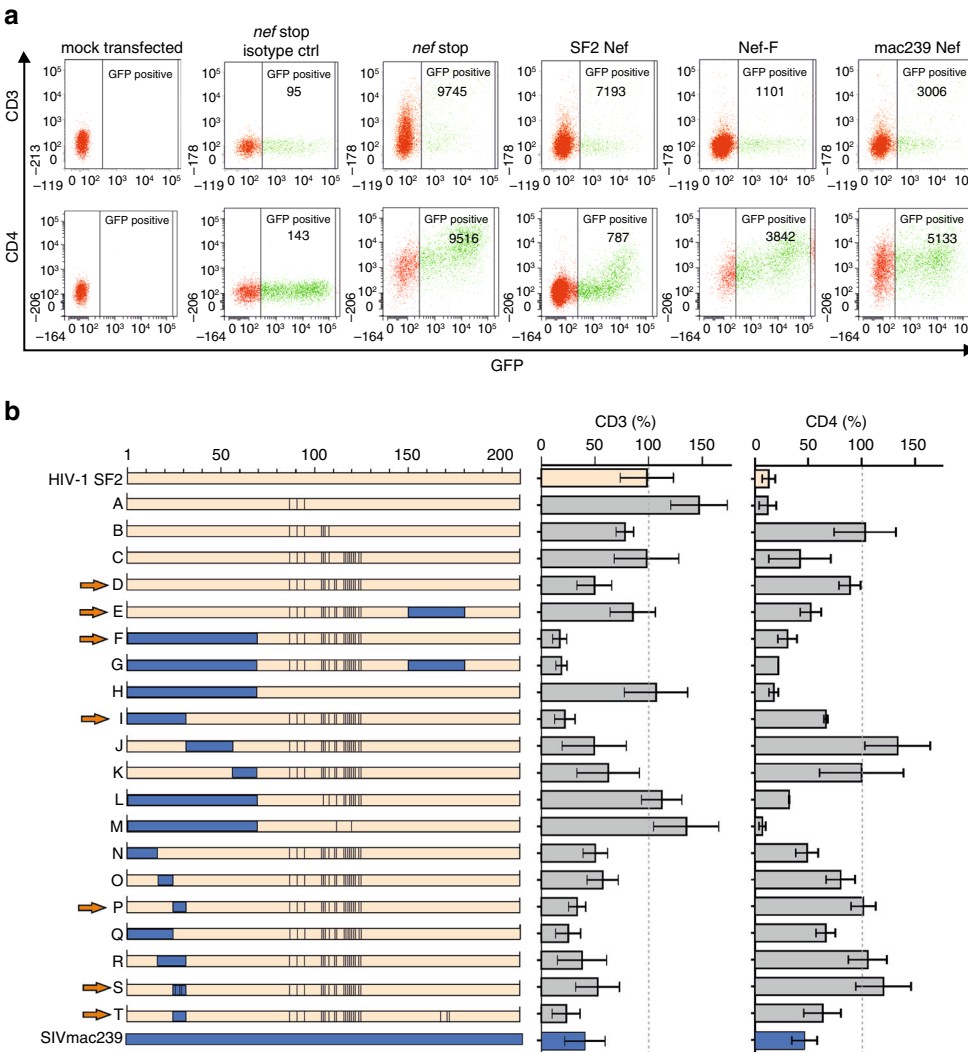

**Fig. 5** Gain-of-function mutations in HIV-1 Nef for CD3 internalization. HEK293T cells were cotransfected with expression vectors for CD3ζ-CD8 or CD4 and constructs coexpressing the indicated *nef* alleles and eGFP via an IRES and examined by flow cytometric analysis 2 days post transfection. **a** Examples for primary FACS data with the gating strategy and the mean fluorescence intensity (MFI) of the stained surface receptor and **b** overview on Nef proteins examined and their CD3 and CD4 downmodulation activities. *Bars* on the left illustrate the composition of the Nef chimeras A–T. Domains derived from HIV-1 SF2 and SIV$_{mac239}$ Nef are highlighted in *beige* and *blue*, respectively. Numbers on *top* indicate the respective amino acid positions. *Red arrows* indicate the Nef proteins that were selected for further analyses in infected primary cells (see Fig. 6). Values on the *right* are normalized to the *nef*-deficient vector control (100%) and represent means of two to four independent experiments (±SEM). Numbers in **a** provide mean fluorescence intensities. See also Supplementary Figs. 5 and 6

receptor internalization by Nef. We aimed at translating this knowledge into the design of an HIV-1 Nef variant capable of CD3 ζ binding and downmodulation. A superimposition of the HIV-1 Nef structure (PDB accession code 3RBB)[38] with the SIV Nef structure determined here reveals the high conformational stability of the backbone residues in the hydrophobic crevice formed by helices α4 and α5 of Nef and the interconnecting loop (Fig. 4a). Within this interaction site, 18 residues vary between the SIV$_{mac239}$ and HIV-1 SF2 Nef sequences, out of which 12 changes are homologous substitutions (Fig. 4b). Based on previous studies[25], we synthesized four different peptides covering either the first or second ITAM motif in CD3 ζ and tested binding to SIV$_{mac239}$ Nef by isothermal titration calorimetry (ITC). Best binding was achieved for a 22-mer peptide encompassing residues 114-135 of CD3 ζ, termed SNID2, containing the second ITAM motif of the immunogenic receptor. A dissociation constant of 2.6 μM was determined for the binding of wild-type SIV Nef to SNID2 (Fig. 4c), while no interaction was seen with the HIV-1 Nef$_{SF2}$ protein (Supplementary Table 1). We exchanged all residues within the

hydrophobic crevice of HIV-1 Nef$_{SF2}$ in nine successive steps to their counterparts in SIV$_{mac239}$ Nef. We also included those residues in the distal part of the two helices α4 and α5 that lie beyond the two gatekeeper residues F94/W117 in HIV-1 or F122/Y145 in SIV Nef, respectively. These large aromatic gatekeeper residues lie in the middle of the two helices and separate the sorting motif recognition site from the RT-loop recognition site of SH3 domains[39]. They also determine the distance of helices α4 and α5 relative to each other and may act like a hinge in the position of the helices. With a $K_d$ of 4.4 μM, best binding was achieved upon a full exchange of all 18 residues in SF2 Nef to SIV$_{mac239}$, termed the H-to-S mutation (Fig. 4d). Binding could also be detected for a 15 and 13 residue exchange mutant (Fig. 4e), whereas fewer mutations did not result in detectable interaction.

**Generation of an HIV-1 Nef gain-of-function mutant for CD3.**
The design of an HIV-1 Nef mutant capable of CD3 ζ binding in vitro prompted us to analyze Nef-mediated receptor

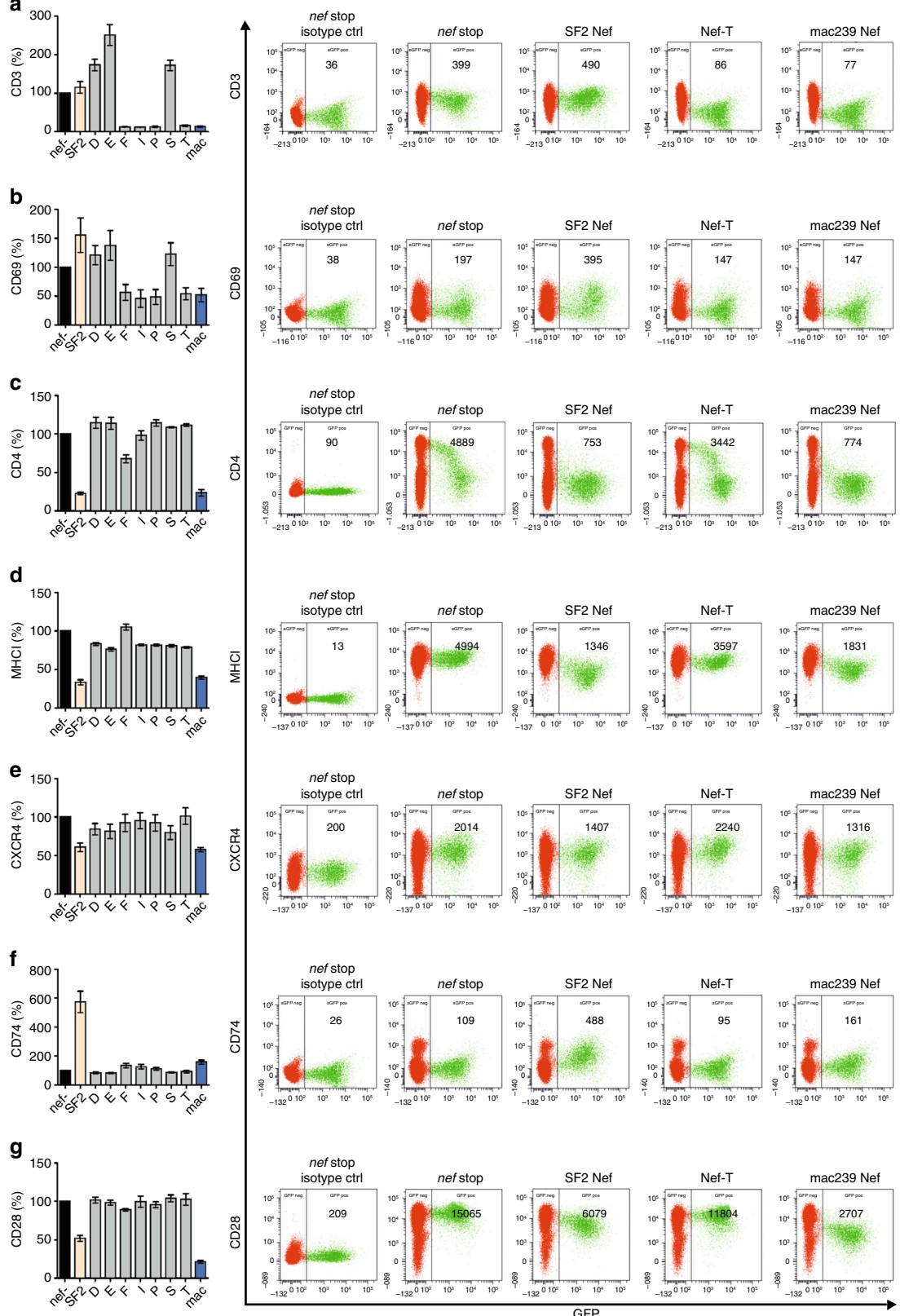

**Fig. 6** Nef-mediated modulation of cell surface receptors in PBMCs. Flow cytometric analysis of **a** CD3, **b** CD69, **c** CD4, **d** MHCI, **e** CXCR4, **f** CD74 and **g** CD28 surface expression in PBMCs infected with VSV-G pseudotyped HIV-1 M NL4-3 IRES eGFP recombinants expressing the indicated *nef* alleles. The eGFP expressing (i.e., infected) population was used to calculate receptor downmodulation. Values were normalized to the *nef*-deficient control (100%). The mean of three independent infections (three donors) ±SEM is shown in the *left column*. Examples for primary data with the gating strategy and the MFI of the stained surface receptor are shown on the *right*

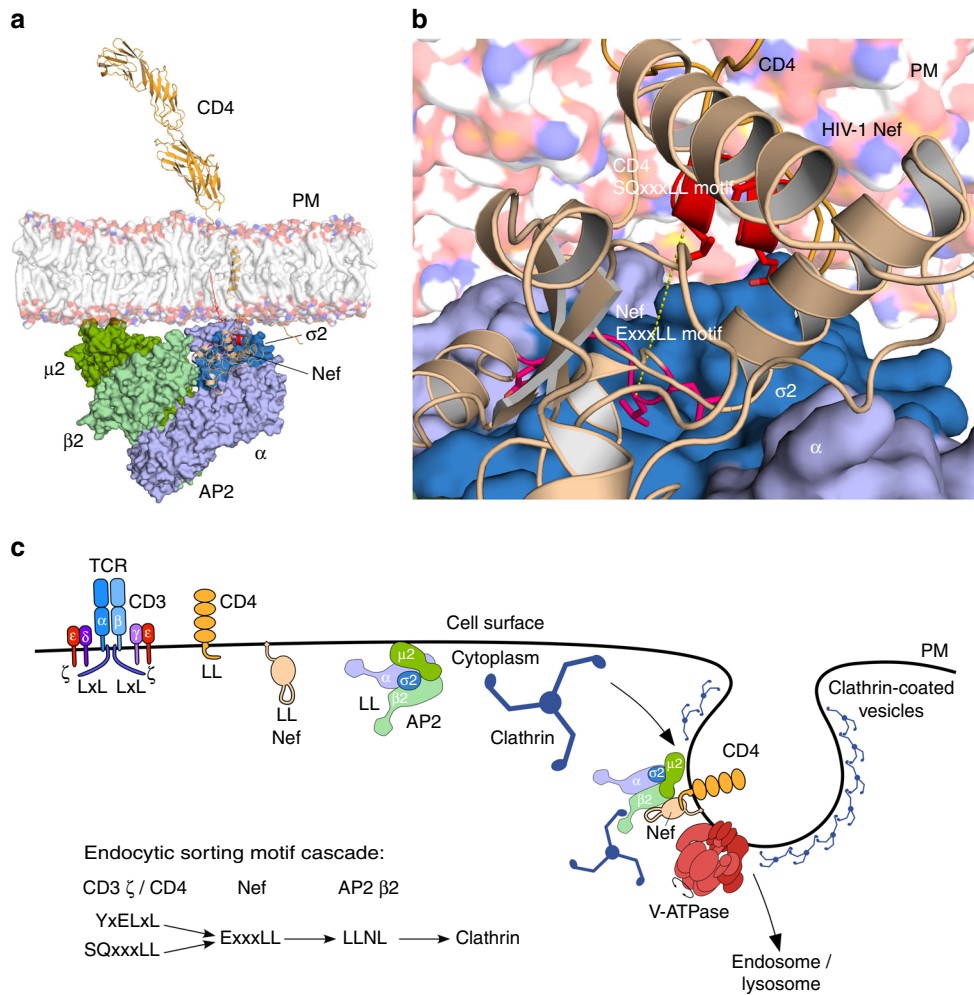

**Fig. 7** Architecture of the CD4/Nef/AP2 complex at the plasma membrane. **a** Structural model of the CD4/Nef/AP2 complex embedded in a dioleoyl model membrane. CD4 (*orange*) contains an extracellular immunoglobulin fold followed by a single transmembrane helix and the C-terminal cytoplasmic domain. Nef (*wheat*) interacts with the SQxxxLL motif of CD4 at the plasma membrane and contacts with the ExxxLL motif in its C-terminal flexible loop the α/σ2 subunits (*blue*) of the hetero-tetrameric AP2 complex. The protein complex is assembled from PDB accession codes 1WIO[46], 2KLU[47], 5NUI (this study), 3RBB[38], 4NEE[43] and 2XA7[44]. **b** Close up of the two dileucine-based sorting motifs of CD4 (SQxxxLL, colored *red*) and Nef (ExxxLL, colored *magenta*) in the model of CD4/Nef/AP2 at the membrane. **c** Nef-mediated endocytosis involves a cascade of dileucine-based sorting motif interactions. Nef acts as an adapter bridging the endocytic sorting motifs of CD3 (YNELNL) and CD4 (SQIKRLL) to the α/σ2 subunits of the AP2 complex. AP2 in turn binds to clathrin via the LLNL motif in the hinge region of its β subunit, thereby inducing the formation of clathrin-coated pits and receptor endocytosis

internalization in cells (Fig. 5a). A set of 20 Nef variants (termed Nef-A to Nef-T) was generated (Fig. 5b), in which specific residues or domains of HIV-1 Nef$_{SF2}$ were exchanged to the corresponding ones of SIV$_{mac239}$ Nef based on the sequence alignment shown in Supplementary Fig. 2. The amino acid sequences of all Nef variants are provided in the Supplementary Note 1. Flow cytometry analyses of *nef* alleles in HEK293T cells that coexpress enhanced green fluorescent protein (eGFP) via an IRES vector confirmed downmodulation of CD3 by SIV$_{mac239}$ but not HIV-1 Nef, whereas CD4 was internalized by both parental Nefs (Fig. 5b). The expression of all Nef constructs was confirmed by Western blotting (Supplementary Fig. 6).

The first four Nef mutants (A–D) successively contained all 18 residues of the hydrophobic crevice that resulted in best binding to the CD3 ζ internalization motif. Nef-D indeed downmodulated CD3, yet it lost its effect on CD4. In the following three variants E, F and G, we exchanged in addition to the 18 mutations of the hydrophobic crevice the entire C-terminal flexible loop, the entire N-terminal membrane anchor domain, or both (Supplementary Fig. 7). The two constructs with the exchanged membrane anchor

domain, Nef-F and Nef-G, enhanced downmodulation of CD3 and CD4 compared to Nef-D. To define the role of the membrane anchor, we tested the effect of the N-terminal amino acids only, Nef-H, and observed full internalization of CD4 but no downregulation of CD3, presumably due to the retrieval of the wild-type HIV-1 hydrophobic crevice.

The Nef variants I–K were designed to define the region within the N-terminal anchor domain responsible for the enhanced CD3 downregulation by Nef-F. Here, the most effective internalization was seen for Nef-I containing the first 37 amino acids of SIV$_{mac239}$. The constructs Nef-L and Nef-M aimed at reducing the number of mutations in the hydrophobic crevice required for CD3 internalization but were inactive (Fig. 5b). We further dissected the N-terminal region enhancing internalization of CD3 ζ by Nef-N to Nef-R, containing three fragment exchanges and two combinations thereof. Analysis of these mutants suggested that region 27–36 of SIV$_{mac239}$ is required for effective internalization of CD3. This stretch in the N-terminal anchor domain of SIV Nef contains a second tyrosine-based sorting motif. In fact, the sequence ETYGRLLGE can be seen as a bona

fide YxxL motif and a cryptic ETxxxLL dileucine-based sorting motif. Accordingly, we mutated the N-terminal YxxL motif and the C-terminal ExxxLL motif to generate Nef-S and Nef-T, respectively. Flow-cytometry analysis showed that the N-terminal YxxL motif but not the ExxxLL motif in the C-terminal flexible loop is required for optimal CD3 internalization (Fig. 5). To test whether the exchange of the N-terminal 10 residues in construct P affects binding of Nef to CD3 ζ, we performed ITC measurements comparing recombinant Nef proteins D and P (Supplementary Fig. 8). Using the same CD3 ζ peptide as before, these Nef proteins exhibited dissociation constants of 8.3 μM (Nef-D) and 4.3 μM (Nef-P), suggesting that not a substantially tighter interaction between Nef and CD3 but rather an enforced interaction with the endocytosis machinery increased CD3 downmodulation activity of the Nef-P variant.

**Mutant HIV-1 Nefs downmodulate TCR-CD3 in primary T cells**. To analyze Nef function in infected primary T cells, we cloned wild-type HIV-1 SF2 and SIV$_{mac239}$ *nef* as well as *nef*-D, *nef*-E, *nef*-F, *nef*-I, *nef*-P, *nef*-S and *nef*-T into an otherwise isogenic HIV-1 NL4-3 construct coexpressing eGFP. The mutant Nefs were selected based on their phenotypes in HEK293T cells. A *nef*-deficient variant served as control and Nef expression was determined by Western blotting (Supplementary Fig. 6). In addition to CD3, CD4 and MHC-I, some Nef proteins also downmodulate CXCR4[40] and CD28[41], and upregulate the MHC-II-associated invariant chain (Ii or CD74)[42]. To obtain more comprehensive information on the functionality of the mutant HIV-1 Nefs, we also analyzed these receptors. Highly effective internalization of CD3 was observed for Nef constructs F, I, P and T, confirming the necessity of the additional N-terminal YxxL motif for this function (Fig. 6a). As expected[11], CD3 levels correlated with the responsiveness of T cells to activation and thus the surface levels of the early activation marker CD69 (Fig. 6b). In contrast, CD4 was only downmodulated upon exchange of the full N-terminal anchor domain, suggesting some crosstalk between the hydrophobic crevice and the membrane anchor domain (Fig. 6c). When narrowing down the sequence requirements in the N-terminus for CD3 internalization (Nef-F, Nef-I, Nef-P), Nef-mediated CD4 downregulation was gradually lost. Similarly, internalization of MHCI, CXCR4, and CD28 and upmodulation of the invariant chain (CD74) that prevents MHC-II-mediated antigen presentation was impaired for all mutant Nefs tested (Fig. 6d–g). Thus, the functional properties of some Nef mutants, i.e., effective downmodulation of TCR-CD3 and suppression of T cell activation, but lack of all other activities, are essentially exactly the opposite of the parental HIV-1 SF2 Nef protein.

**Architecture of the CD4/Nef/AP2 complex**. The crystal structure of Nef binding to an ExxxLM motif enables us to build a full CD4/Nef complex structure combining structural data from previously published studies[43–47]. The CD4/Nef assembly can be extended to the AP2 complex, as a complex structure of Nef binding to a truncated α/σ2 hemi-complex has been reported[43]. This structure can be combined with the tetrameric AP2 trunk complex[44] including the inositol phosphate binding sites of the α and μ2 subunits that determine the association to the plasma membrane[45]. A model of the multiple structures assembled into the full architecture of the trimeric CD4/Nef/AP2 complex embedded in a plasma membrane leaflet is shown in Fig. 7a. The architecture of the full CD4/Nef/AP2 complex reveals how Nef interacts with CD4 directly at the plasma membrane, while the interaction of the dileucine recognition site in AP2 with the dileucine motif of Nef takes place 12 Å downstream of the membrane plane in the cytosol (Fig. 7b). HIV-1 Nef is thus

able to bridge the distance between the cytoplasmic tail of CD4 that is embedded in the membrane in the absence of a serine phosphorylation and couple CD4 to AP2 by exposing its own dileucine motif into the cytosol. In this way, Nef functions as an adapter between the SQxxxLL dileucine motif of CD4 and the dileucine motif recognition site of AP2 by recruiting CD4 at the plasma membrane. How the CD4/Nef/AP2 complex assembles with clathrin via the LLNL motif in the β2 subunit of AP2 to form clathrin-coated pits is not clear at present (Fig. 7c). Clustering of CD4/Nef and the induction of positive membrane curvature might be one mechanism for Nef-stimulated clathrin-mediated endocytosis.

## Discussion

In this study, we analyzed how lentiviral Nef proteins interact with dileucine-based sorting motifs to promote internalization of the TCRs CD4 and CD3 from the surface of infected cells. We find that Nef is indeed able to act like an adapter by binding to dileucine-based sorting motifs embedded in the plasma membrane while simultaneously exposing its own dileucine motif within the C-terminal flexible loop to the cytosol. This is possible by a dipolar electrostatic surface charge character of Nef. The hydrophobic crevice that interacts with the dileucine motifs is surrounded by positively charged residues, whereas the C-terminal flexible loop with the ExxxLL sorting motif in its center is largely negatively charged. This enables Nef to pick up the cytosolic tail sequences of T cell surface receptors at the plasma membrane and couple them to the endocytic adapter protein machinery. Strikingly, the same surface of Nef interacts with SQxxxLL and YxELxL-based sorting motifs as contained in the cytoplasmic tails of CD4 and CD3, respectively. However, while both motifs are part of helical structures, the binding angle relative to the recognition site in the hydrophobic crevice of Nef differs by almost 50°, leaving only the central leucine at the same position.

On a molecular level, the structure of Nef bound to the ExxxLM motif of another Nef molecule allowed us to determine the contribution of individual residues to this direct interaction. Mutation of the conserved double arginine motif R109 and R110 in HIV-1 Nef$_{NL4-3}$ to alanine has long been known to abrogate CD4 downregulation[36, 37]. Here, we show that these two residues at the beginning of helix α5 form a specific salt bridge with the glutamate of the ExxxLM motif, whose electrostatic interaction might help to attract the sorting motif to its recognition domain. The two hydrophobic moieties Lφ instead interact with the hydrophobic crevice in the core domain of Nef. Eight residues constitute this hydrophobic crevice at the surface of Nef that is flanked by the two gate keeper residues W117 and F94 in HIV-1 Nef and demarcate the sorting motif recognition site from the binding site for SH3 domains. The Nef–ExxxLL complex structure confirms previous mapping data from NMR chemical shift perturbation experiments showing that L97 in Nef$_{NL4-3}$, corresponding to L129 in SIV$_{mac239}$ Nef, directly interacts with the dileucine motif of CD4[29]. The hydrophobic crevice is surrounded by basic residues K96, K98, R109, R110 in HIV-1 Nef$_{SF2}$ or K124, K126, R137, R138 in SIV$_{mac239}$ Nef, providing a positively charged surface to this interaction site. In contrast, the rear side to this crevice on the surface of Nef and the C-terminal flexible loop are highly negatively charged[38], suggesting that the hydrophobic crevice faces the negatively charged plasma membrane when bound to the lipid bilayer. In agreement with our model, not only mutations of R109 and R110, but also hydrophobic residues within the binding pocket have previously been shown to abrogate the ability of HIV-1 Nef to decrease CD4 surface levels[48, 49]. Intriguingly, FRET experiments

demonstrated that mutation of Leu114, which interacts with the second leucine residue of the ExxxLL motif, fully disrupts the interaction of HIV-1 Nef with CD4[48]. Furthermore, we here expand these functional analyses to CD3 and analyzed mutations of R137 and R138 in SIVmac, SIVsmm, and HIV-2 Nef. Our finding that the respective mutants lose their ability to downmodulate CD3 and CD4, but not MHC-I, supports an important role of the interaction of these two residues with the acidic residue in the ExxxLL motif. Overall, the hydrophobic crevice constitutes a characteristic binding site in Nef for the interaction with endocytic sorting motifs that might well be suitable for targeting by small molecular compounds.

The comparison of SIV Nef structures interacting with endocytic sorting motifs contained in CD4 or CD3 ζ served as a blueprint for the rational development of an HIV-1 Nef variant capable of CD3 downregulation. Notably, not only amino acid changes in the hydrophobic crevice of HIV-1 Nef enhancing the affinity for the CD3 ζ chain in vitro but also an N-terminal tyrosine-based sorting motif were required for efficient Nef-mediated downmodulation of CD3 from the cell surface. In agreement with data on SIV Nef[41], the ExxxLL sorting motif in the C-terminal flexible loop of Nef instead was dispensable for CD3 internalization. While both effects, Nef-mediated CD3 and CD4 downregulation, require AP2 and clathrin-mediated endocytosis, we speculate that a different accessibility of Nef in the lumen upon binding to these receptors may lead to the requirement of the sorting motif in the N-terminal membrane-targeting domain. Binding to the large multi-subunit CD3 TCR could thus favor different conformations of Nef compared to binding to the 37 residues encompassing cytoplasmic tail of CD4. As a consequence these quaternary assemblies might influence the interaction to AP2. Instead of binding to the α/σ2 subunits of AP2, as found for the endocytic CD4/Nef complex, CD3-bound Nef may recruit AP2 via binding of the N-terminal YxxL motif to the μ2 subunit due to sterical constraints.

In contrast to CD3 downmodulation, internalization of CD4 does not depend on the presence of a tyrosine-based sorting motif in the N-terminus of Nef. Our modeled assembly of the endocytic CD4/Nef/AP2 complex reveals that Nef acts as an adapter for CD4 and AP2 as its recognition site for dileucine-based sorting motifs is placed about 12 Å closer to the plasma membrane than the corresponding interaction site in the α/σ2 subunits of the AP2 complex. Nef might therefore be able to interact with the dileucine-based sorting motif of CD4 in the lipid–water interface of the plasma membrane, whereas in the absence of Nef CD4 needs to be phosphorylated at S433 in order to be exposed from the membrane leaflet and recognized by AP2 for internalization[22]. The kinetics of the assembly of the tripartite CD4/Nef/AP2 complex, however, remains enigmatic. From a biochemical perspective it is surprising that Nef, containing itself a bona fide dileucine-based sorting motif, does not bind to AP2 in the absence of cargo uptake. Instead of stimulating receptor internalization this could lead to a saturation of the AP2 dependent internalization machinery. The absence of such saturation mechanism suggests a well-controlled order of binding events. Notably, Nef is not always localized to the plasma membrane but stays in the cytosol most of the time[50]. When binding to the plasma membrane via its N-terminal myristate the own dileucine motif might be hidden, e.g., by an autoregulatory mechanism, whereas it becomes exposed upon binding to a receptor internalization motif. Such interactions might lead to exposure of the Nef internalization motif, which is subsequently recognized by AP2. This would secure that Nef could only interact with AP2 when cargo is taken up. Using fluorescence-based kinetics we previously observed conformational changes of Nef upon association with membranes and insertion of the myristate into

the lipid bilayer[51]. A transition from a closed-to-open conformation upon membrane association was also seen by neutron and X-ray reflection methods[52]. Such autoregulatory-driven conformational changes in Nef may induce a cascade of dileucine-based sorting motif interactions that could be triggered upon membrane association and cargo uptake. This could also include the binding of the LLNL motif in the hinge region of the AP2 β subunit to clathrin, which ultimately leads to formation of clathrin-coated pits and endocytosis (Fig. 7).

In summary, our structural model of a hydrophobic crevice in HIV-1 Nef able to target both, dileucine-based and tyrosine-based sorting motifs, helps to explain how this multifunctional viral protein can downmodulate the expression of multiple surface receptors, including CD4 and CD3. Furthermore, the hydrophobic crevice is a striking example of how primate lentiviral proteins adapt specific domains to target diverse cellular proteins. Finally, our structural analyses of Nef allow the generation of gain-of-function or loss-of-function mutants that will not only be useful tools to investigate the importance of individual Nef functions for viral replication but also reveal whether the hydrophobic crevice may be a potential therapeutic target.

## Methods

**Plasmids.** The coding sequence of SIV$_{mac239}$ Nef (GenBank accession number AY588946.1) was modified at nucleotide position 8449 to G in order to change from an ocher codon to glutamate 93 in Nef. For recombinant protein expression, protein coding genes were amplified from a cDNA plasmid codon-optimized for expression in *Escherichia coli* with primers containing *Nco*I and *Eco*RI restriction sites at the 5′ and 3′ ends, respectively. The oligonucleotide primers used in this study are listed in Supplementary Table 2. PCR products were cloned into the pGEX-4T1 (Amersham Bioscience) expression vector modified with a TEV protease cleavage site. Multiple expression constructs with various domain boundaries were generated as GST-fusion proteins with the protein products 66–235 (SIV Nef-B) and 87–235 (SIV Nef-E) being most stable and showing the least aggregation behavior. For in vitro measurements of CD3 ζ binding, HIV-1 Nef gain-of-function mutations (termed HIV Nef-A to Nef-T) were cloned as His-tagged proteins with domain boundaries 45–210 in the pProEx-HTa expression vector. In addition, Nef-D and Nef-P mutants were cloned with domain boundaries 23–210 for ITC experiments. The expression plasmid for the SH3 domain of human Hck (GenBank accession number NM_001172129), residues 79–138, was generated as described[38]. A mutation was introduced in the RT-loop sequence of Hck, termed Hck$_{SH3-E}$, that modified the wild-type sequence E$_{90}$AIHHE of six residues to the five amino acids E$_{90}$GWWG for enhanced binding to SIV Nef[31].

**Protein expression and purification.** SIV and HIV-1 Nef proteins were expressed as N-terminal GST-fusion proteins in *E. coli* BL21(DE3) cells (New England Biolabs, Inc.) and purified by affinity chromatography. For crystallization purposes, Hck$_{SH3-E}$ (residues M78-79-138) was coexpressed from a pET-28b vector without any affinity tag. Bacteria were cultured in LB medium containing 100 μg ml$^{-1}$ ampicillin and 50 μg ml$^{-1}$ kanamycin at 30 °C and induced at OD$_{600}$ = 0.8 with 0.5 mM IPTG for additional 5 h. Harvested *E. coli* cells were resuspended in 20 mM HEPES (pH 8.0), 500 mM NaCl, 1 mM TCEP, 1 mM PMSF, lysed by fluidizer and cleared by centrifugation at 30,000 *g* for 45 min. SIV$_{mac239}$ Nef-B alone or co-purified with Hck$_{SH3-E}$ and SIV$_{mac239}$ Nef-E alone were separated from *E. coli* proteins by affinity chromatography using GSH resin (Amersham Bioscience) and then size exclusion chromatography on a 16/60 Superdex 75 column (Amersham Bioscience). Purified proteins were analyzed by SDS PAGE and mass spectrometry and stored in 20 mM HEPES (pH 8.0), 100 mM NaCl, 1 mM TCEP at −80 °C.

The CD3 ζ SNID2 peptide (UniProt accession number sequence: P20963, aa 115–135; A-QKDKMAEA**YSEIGM**KGERRRG), comprising SNID2 and partly overlapping with the ITAM 2, was synthesized in house by Boc-chemistry. After deprotection and cleavage from the resin with water-free HF and p-cresol as scavenger for 1 h at 0 °C, the peptide was purified by RP-HPLC on a preparative C4 or diphenyl column using linear gradients from buffer A (0.1% (v/v) TFA in water) to B (0.08% (v/v) TFA in ACN). ESI-MS and analytical HPLC were used to identify product-containing fractions for lyophilization. In addition, three CD3 ζ peptides SNID1 (QL**Y**N**EL**N**L**GR; aa 70–79), SNID2-s (EA**YSEIGM**KG; aa 121–130) and SNID2-m (EA**YSEIGM**KGERRRG; aa 121–135), as well as the two CD4 peptides CD4$_{LL1}$ (AYQ-QAERM**SQIKRLL**S; aa 428–440) and CD4$_{LL2}$ (ERM**SQIKRLL**SEKKT-Y; aa 430–444), both equipped with an additional tyrosine residue for accurate concentration determination, were purchased from Biosyntan, Berlin.

**Crystallization of Nef–Hck$_{SH3}$ and Nef–CD4$_{LL1}$ complexes.** Crystals of the Nef-B–Hck$_{SH3-E}$ complex were grown at room temperature by mixing equal volumes of

protein complex (9 mg ml$^{-1}$) and reservoir solution (12% PEG 3350, 0.15 M tri-lithium-citrat, 1% 1.6 hexandiol) using the hanging drop vapor diffusion method. For cryo-protection, crystals were transferred to reservoir solution supplemented with 20% ethylene glycol, loop mounted, and flash frozen in liquid nitrogen.

A synthetic peptide comprising the cytosolic dileucine motif of CD4 (peptide CD4$_{LL1}$) was added to recombinant Nef-E in 2.5-fold molar excess in attempt to form a Nef-E–CD4$_{LL1}$ complex. Initial crystallization conditions were identified by adding 0.1 µl of the Nef-E–CD4$_{LL1}$ sample (10 and 5 mg ml$^{-1}$) to 0.1 µl reservoir solution on 96-well Hampton 3553 sitting-drop plates (Greiner) at 277 K. Conditions consisting of 15% PEG 4000, 0.15 M ammoniumsulfat, 0.1 M MES (pH 6.0), and 5 mg ml$^{-1}$ protein concentration yielded weakly diffracting crystals. Conditions were optimized to 10% PEG 4000, 0.15 M ammonium sulfate, and 0.1 M MES (pH 5.8) at same protein concentration, using the hanging-drop method in 24-well Linbro plates. Crystals grew within 3 days to a size of $100 \times 50 \times 40$ µm. Prior to flash-cooling in liquid nitrogen, the crystal was incubated for 15 s in a cryoprotectant solution containing the mother liquor components and additional 20% ethylene glycol.

**Data collection and processing.** Native diffraction data sets of all crystals were recorded at cryogenic temperature of 100 K using a PILATUS 6 M detector on beamline X10SA (PXIII) at the Swiss Light Source (SLS, Villigen, Switzerland). The data was processed, integrated, and scaled using the XDS package[53]. Data collection and refinement statistics are summarized in Table 1.

The space group of the Nef-B–Hck$_{SH3-E}$ crystal was identified as $P3_2$ with unit cell dimensions of $a = b = 104.0$, $c = 53.0$. The asymmetric unit contained two copies of Nef-B–Hck$_{SH3-E}$. The structure was solved by molecular replacement with phaser[54] using the previously published SIV Nef structure as the search model (PDB: 3IK5)[35]. After initial simulated annealing (torsion angle) in phenix.refine[55], the model was refined to a resolution of 2.78 Å by multiple iterations of alternating manual model rebuilding in COOT[56] and positional refinement with phenix. refine[55] using non-crystallographic symmetry restraints.

The crystal grown of Nef-E–CD4$_{LL1}$ diffracted up to 2.50 Å. The space group was identified as $C222_1$ with unit cell dimensions of $a = 51.48$, $b = 140.73$, $c = 106.43$, which gave enough space for two Nef-E molecules. A 1.35 Å SIV Nef-core structure, which was previously solved in our group resolving SIV$_{mac239}$ Nef residues 103–183, 204–233 was used as a search model in PHASER (CCP4 package)[57] to solve the Nef-E–CD4$_{LL1}$ structure with molecular replacement. Alternating cycles of manual building in COOT[56] and refinement in REFMAC5[58] and Phenix[55] were performed to generate the final structural model. Data collection statistics and refinement parameters for both structures are shown in Table 1. Molecular diagrams were created using PyMOL (Schrödinger, MA). The electrostatic surface potential was calculated with adaptive Poisson-Boltzmann solver (APBS)[59].

**Isothermal titration calorimetry.** Interactions of SIV$_{mac239}$ Nef or HIV-1 Nef$_{SF2}$ proteins with various fragments of the CD3 ζ ITAM region were analyzed by ITC using a MicroCal iTC200 microcalorimeter (Malvern, UK). Measurements were done in 20 mM HEPES buffer (pH 8.0), 100 mM NaCl, at 25 °C. Typically, Nef at a concentration of 200 µM was injected stepwise from the syringe to 20 µM CD3 ζ placed in the measurement cell. The change in heating power was monitored over the reaction time until equilibrium was reached. Data were analyzed using the software provided by the manufacturer.

**Size exclusion chromatography.** Analytical gel filtration experiments of recombinant SIV$_{mac239}$ Nef, Hck$_{SH3-E}$, and CD3 ζ fragments were performed with a multicomponent Waters 626 LC system (Waters, MA) connected to a Superdex S75 (10/300 GL) column (GE Healthcare). Typically, 100 µl of a 1.5 mg ml$^{-1}$ protein solution was injected onto the column that was previously equilibrated with 10 mM Tris/HCl (pH 9.0), 100 mM NaCl buffer. Analytical gel filtration runs were done at a flow rate of 0.5 ml per minute at 293 K. The optical density was monitored at a wavelength of 280 nm. Gel filtration experiments were performed repeated times.

**Mammalian expression plasmids.** Bi-cistronic CMV-promoter-based pCG expression vectors coexpressing *nef* and the eGFP have been described previously[60]. Briefly, splice overlap extension PCR with primers introducing *Xba*I and *Mlu*I restriction sites flanking the reading frames was used to generate chimeric *nef* alleles. PCR fragments were then purified from agarose gels and inserted into the pCG vector using standard cloning techniques. Human CD4 was expressed from a previously described pCDNA3.1(+)-derived plasmid[60]. To express the CD3 ζ chain, CD8 was fused to the cytoplasmic domain of CD247 and cloned into the pCG expression vector via *Xba*I/*Mlu*I as described[61]. All plasmids were sequenced to confirm their accuracy.

**Proviral constructs.** Generation of the HIV-1 NL4-3-based proviral constructs carrying functional the *nef* genes followed by an IRES and the eGFP gene has been described[11]. Briefly, splice overlap extension PCR was performed to replace the HIV-1 NL4-3 *nef* allele with the *nef*s of HIV-1 M SF2, SIV$_{mac239}$ or chimeras thereof. The proviral HIV-1 constructs are replication competent and express all viral genes, including *nef*, through the regular LTR promoter and splice sites.

The integrity of all PCR-derived inserts was confirmed by DNA sequencing. *nef*-defective control constructs contain premature stop codons.

**Cell culture.** HEK293T and Jurkat T cells were obtained from ATCC. HEK293T cells were maintained in Dulbecco's Modified Eagle Medium supplemented with 10% FCS, penicillin (100 U ml$^{-1}$), streptomycin (100 µg ml$^{-1}$), and L-glutamine (2 mM). Jurkat T cells were cultured in RPMI1640 medium with 10% FCS, penicillin (100 U ml$^{-1}$), streptomycin (100 µg ml$^{-1}$), and L-glutamine (2 mM). PBMCs from healthy human donors were isolated using lymphocyte separation medium (Biocoll Separating Solution, Biochrom), stimulated for 3 days with PHA (phytohaemagglutinin; 1 µg ml$^{-1}$), and cultured in RPMI1640 medium with 10% FCS, penicillin (100 U ml$^{-1}$), streptomycin (100 µg ml$^{-1}$), L-glutamine (2 mM), and 10 ng ml$^{-1}$ IL-2 prior to infection.

**Western blotting.** HEK293T cells were harvested 48 h post transfection and PBMCs were harvested 72 h post transduction by lysing them in Western Blot lysis buffer (150 mM NaCl, 50 mM HEPES, 5 mM EDTA, 0.1% (v/v) NP-40, 0.5 mM Na$_3$VO$_4$, 0.5 mM NaF, pH 7.5). After addition of Protein Loading Dye (LI-COR) and 2.5% β-mercaptoethanol, samples were incubated at 95 °C for 5 min, separated on a NuPAGE Bis-tris 4–12% gradient gel (Invitrogen) and blotted onto an Immobilon-FL PVDF membrane (Merck Millipore). The membrane was blocked in 5% milk and proteins were stained using primary antibodies directed against HIV-1 Nef (NIH #2949; dilution 1:200), GAPDH (BioLegend #631401; dilution 1:200) and HIV-1 p24 (abcam #ab9071) and secondary antibodies conjugated with Infrared Dyes (LI-COR). Bands were detected using the infrared imager Odyssey9120 (LI-COR) and the Image Studio Lite Version 4.0 (LI COR).

**Flow cytometric analysis.** To determine the effect of Nef on CD4 and CD3, HEK293T cells were transfected by the calcium phosphate method with 5 µg of an expression vector for *nef* or the respective vector control expressing only eGFP and either 1 µg of an expression plasmid for CD4 or 0.2 µg of an expression plasmid for CD3ζ–CD8. Two days post transfection, cells were stained for CD4 (MHCD0405, Life Technologies) or CD8 (555369, BD) and analyzed by two-color flow cytometry. Cell surface receptor downmodulation was calculated as described[60]. Briefly, the mean fluorescence intensities were determined for cells showing specific ranges of GFP expression. Isotype control antibodies were used to determine background fluorescence and unspecific antibody binding. The mean fluorescence intensities obtained with the isotype control antibodies were subtracted from those obtained with specific surface receptor antibodies. The fluorescence values obtained for cells transfected with the control construct expressing only GFP was compared with the corresponding value obtained for cells coexpressing Nef and GFP. The same regions of GFP expression were used in all calculations.

To analyze modulation of CD3, CD4, and MHC-I in T cells, Jurkats were transfected with bicistronic vectors coexpressing Nef and eGFP using the DMRIE-C reagent (Gibco-BRL), according to the manufacturer's instructions. To increase transfection rates, Jurkat T cells were stimulated with PHA[62]. For quantification of Nef-mediated receptor downregulation, the respective mean fluorescence intensities were determined for cells expressing no, low, medium, or high levels of GFP. To calculate the specific effect of Nef on receptor cell surface expression, the mean fluorescence intensities obtained for cells coexpressing Nef and GFP were normalized to those obtained with the *nef*\* construct expressing GFP only.

To analyze cell surface receptor modulation in primary cells, PBMCs were infected with VSV-G-pseudotyped HIV-1 NL4-3 IRES eGFP expressing different *nef* alleles. Three days post infection, cells were stained for CD4 (MHCD0405, Life Technologies), CD3 (555333, BD), MHCI (R7000, Dako), CD28 (348047, BD), CD74 (226-050, Ancell), or CXCR4 (555976, BD) and analyzed by flow cytometry. To analyze T cell activation, cells were stimulated with PHA 3 days post infection. One day later, cell surface CD69 (555531, BD) was stained and analyzed by fluorescence-activated cell sorting (FACS). Modulation of all surface receptors was calculated as described above for HEK293T cells.

**Data availability.** Atomic coordinates and structure factors for SIV$_{mac239}$ Nef and the SIV$_{mac239}$ Nef–Hck$_{SH3}$ complex have been deposited in the Protein Data Bank with accession codes 5NUI and 5NUH, respectively. All other data supporting the findings of this study are available from the corresponding author upon reasonable request.

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

## Acknowledgements

We thank Diana Ludwig, Karin Vogel-Bachmayr, Susanne Engelhart, Daniela Krnavek, and Kerstin Regensburger for excellent technical assistance, Steffen Wildum for providing some of the flow cytometry data, and Sonja Kühn and Ingrid Vetter for advice on crystal data analysis. This work was supported by a grant from the Deutsche Forschungsgemeinschaft (DFG) to M.G. (GE 976/6-1). M.G. is a member of the DFG excellence cluster ImmunoSensation. F.K. is supported by European FP7 "HIT HIDDEN HIV" (305762), DFG CRC 1279 and SPP 1923, and an Advanced ERC investigator grant. D.S. is supported by the DFG Priority Programme SPP 1923.

## Author contributions

S.M. and F.A.H. expressed, purified, and crystallized proteins and together with K.A. determined the crystal structures. D.S., H.Y. and D.H. performed cell-based assays and analyzed data. S.L. performed ITC experiments and complex modeling. M.G. conceived the study, F.K. and M.G. supervised the research, and D.S., F.K. and M.G. wrote the manuscript with input and editing from all the authors.

## Additional information

**Competing interests:** The authors declare no competing financial interests.

