## [Peer Review file · Nature Communications]

Reviewers' comments:

Reviewer #1 (Remarks to the Author):

Lentiviral Nef proteins interact with and modulate various cellular receptors, to facilitate viral replication and interfere with immune responses. Some cellular receptors, such as CD4, are targeted by both SIV and HIV Nef proteins, whereas other, such as CD3, are modulated by SIV and HIV-2 Nef but not by HIV-1. The structural bases for these differences are poorly characterized. Nef proteins are known to form oligomers. Manrique and colleagues solved the structure of SIV Nef bound to the ExxxLM motif of another Nef protein. They also reconstituted how Nef-CD4-AP2 complexes may be formed. Furthermore, by comparing SIV and HIV-1 Nef structures, they determined the structural elements necessary for the down-modulation of CD3. This structural analysis was combined to a functional study of a panel of HIV-1 Nef mutants. The activity of the mutants on various cellular receptors was analyzed when Nef was expressed alone, or within a full virus during infection of PBMC. This led to the characterization of key residues required for the activity of lentiviral Nef.

I am unable to assess the structural analysis part because this is not my area of expertise. However, experiments are well performed and convincingly presented. It is clearly shown how the introduction of mutations in HIV-1 Nef rescues the CD3 down-modulation of this protein. This work provides new structural insights into how Nef proteins interact with different cellular receptors. I don't have major concerns.

Minor comments

1. There are some minor discrepancies in the results obtained when Nef mutants are expressed as a single protein or from a proviral construct. For instance, Nef mutants D and S down-modulate CD3 when expressed alone, and not in infected cells. What are the overall amounts or stability of the mutants in the two experimental systems?
2. HIV-1 Nef does not down-regulate CD3, but impairs the formation of the immune synapse, indicating that HIV-1 may have evolved other means to interfere with T cell activation. This could be discussed.
3. The authors could discuss further the structural differences between SIV and HIV-2 Nef structures.

Reviewer #2 (Remarks to the Author):

This work reports new X-ray crystal structures of an SIV mac239 Nef core protein alone and in complex with a modified human Hck SH3 domain. While the authors included the human CD4 cytoplasmic tail peptide, containing the ExxxLL endocytic motif required for Nef binding, no electron density was found for the CD4 peptide in these structures.

SIV Nef packed in the crystal with the C-terminal flexible loop helix of one Nef monomer making contacts with alpha helices 4 and 5 of an adjacent Nef in the trimer. Since this SIV Nef loop (and by extension the analogous HIV-1 Nef loop) houses an endocytic di-leucine motif, this interaction is used as a model for the endocytic ExxxLL motif recognition site for CD4, for which there is currently no structure for HIV-1 Nef. The hydrophobic pocket site on SIV Nef where the loop interacts is also the site where the CD3 zeta chain peptide interacts with a peptide dileucine motif as shown in a previous crystal structure of the SIV Nef mac239 core in complex with a CD3 tail peptide.

No HIV-1 Nef structure in complex with the CD4 tail peptide has been determined, and insight into that structure is important. However, the novelty of these SIV Nef X-ray structures is reduced by

not having the CD4 peptide bound. The model based on the Nef:Nef interaction offers some insight, but doesn't really extend recent work by the Hurley group in terms of what the Nef:CD4:AP2 complex may look like at the plasma membrane (Ren et. al eLife 2014; citation below). Also HIV-1 Nef does not normally induce downregulation of CD3 zeta – thus the value of reconstituting this function by extensive mutagenesis as presented in the paper does not provide new insight regarding HIV-1 Nef function.

Specific Points:

1. Structural alignment and helical wheel analysis of Nef, CD3 and CD4 di-leucine motifs with Nef hydrophobic pockets suggests important residue contact information. Mutagenesis of new contact residues in the SIV and HIV-1 Nef hydrophobic pockets revealed in these structures should be performed to validate CD4 and CD3 di-leucine peptide binding, and extended to functional analyses in terms of receptor downregulation.
2. It is not clear why the authors cut off the X-ray data at 2.6 angstroms for the Nef core structure since mean I/σ in the highest resolution shell is 3.6; the X-ray data are likely to be useful to higher resolution. It is common practice to cutoff x-ray data to an I/σ of 2 or 1.5.
3. The difference in the R_{work}/R_{free} refinement values of 7% is unacceptable for structure refinement and may be an indication of model bias. The authors need to improve the structure refinement by including refinement using simulated annealed omit maps.
4. The number of residues with ϕ/ψ angles in the allowed region of the Ramachandran plot is much too large. This is another indication that the structure needs to be better refined.
5. A few aspects of the flow cytometry data require further clarification (Figure 5A and 6). What are the 'control' plots – presumably these cells are uninfected and not stained with antibodies. This should be explicitly indicated. In addition, a second (or alternative) control of uninfected cells stained with both antibodies should be included for comparison. The gating strategy is not completely clear especially along the y (receptor) axis. In Figure 5A, none of the panels seem to show a significant signal with the CD3 antibody. Where is the cutoff to determine the number CD3-positive cells? The same issue is present for Figure 5a with CD4. Also the number of CD4-positive cells seems to increase in the presence of increasing amounts of HIV-1 SF2 Nef, whereas the opposite would make more sense. Having the uninfected/double-stained controls should help here. Cut-offs for each flow panel in Figure 6 should also be added; contour plots may help as well.

Reference cited: Ren, X.; Park, S. Y.; Bonifacino, J. S.; Hurley, J. H. (2014) How HIV-1 Nef hijacks the AP-2 clathrin adaptor to downregulate CD4. *Elife*. 3, e01754.

Reviewer #3 (Remarks to the Author):

NCOMMS-16-29131-T

The Nef protein of lentivirus facilitates internalization of CD3, CD4 or other immune receptors by recruitment of AP complex to induce clathrin-dependent endocytosis. Internalization of CD3 or CD4 in regular situations (i.e., without the Nef protein) is coupled with phosphorylation of their intracellular regions, whereas their Nef-mediated internalizations do not require such phosphorylation. The Nef proteins can recognize non-phosphorylated internalization motifs and also interact with the AP complex through their own dileucine-based sorting motifs. The Nef proteins can act as adaptors linking the AP complex and immune receptors. The internalization of CD4 mediated by the Nef proteins of HIVs and SIVs may promote the release of fully infectious virus particles and prevent super-infection. On the other hand, CD3 can be internalized by the Nef proteins of HIV-2 and most SIVs but not by that of HIV-1. The recognition manner of the tyrosine-based sorting motif of CD3 by SIV Nef has been reported, whereas that of the dileucine-based motif of CD4 has remained unclear. In this study, Manrique et al. aimed to reveal mechanisms for

recognition of the dileucine-based sorting motif of CD4 by Nefs of HIV-1 and SIV. However, they failed to crystalize the complex between CD4 and HIV-1 or SIV Nef. Instead, they obtained the SIV Nef structure, where the dileucine-based sorting motif of one SIV Nef (for the interaction with the AP2 complex) is recognized by another adjacent SIV Nef in the crystal. The dileucine-based motif of SIV Nef (ED'E'EHY'LM'HP) may mimic that of CD4 (MS'Q'ILR'LL'SE). This idea aided their subsequent mutational analyses to investigate how the tyrosine-based motif of CD3 and the dileucine-based motif of CD4 are discriminated, together with the previously reported complex structure between SIV Nef and CD3 ζ .

The crystallography and binding experiments using ITC appears to be properly done. The cell-based internalization assays may be consistent with the structure and results of the binding experiments. However, hydrophobic-interaction-based binding might be promiscuous and interacting residue pairs can be changed in some cases. I am not confident that the recognition of 'EEHYLM' from SIV Nef is (basically) identical to that of 'QILRLL' from CD4. In addition, the N-terminal YxxL motif of SIV Nef makes the interpretation of the results from the internalization assay complicated. Overall, the key mechanism for discrimination between the tyrosine- and dileucine-based sorting motifs by Nef is not clear. The findings in this study seem to be of interest in the limited field.

Finally, I would like to suggest several points to be improved as follows:

1. Hexagonal or trigonal assembly of Nef in the crystal may be artifact in crystallization, if there is no finding suggesting that similar assemblies occur in cells or at least in solution. Fig. 1a and b should be removed or moved to Supplementary information. In Fig. 1d, a type of electron density map should be described (probably 2Fo-Fc map).
2. In Fig. 2, I prefer stick and cartoon models for presentation of protein-protein interactions. Electrostatic surface is not so informative. Fig. 2c could be moved to Supplementary information.
3. In Fig. 4b, mutated residues are not easily recognized. The SIV and HIV residues may be presented in different colors.
4. In the left panel of Fig. 5, explanation of mutants 'A-T', color bars and red arrows should be added in the figure legend.
5. The letters representing cell constants (a, b, c, alpha, beta, gamma) and the initial letter of the space group should be italicized.
6. PDB codes are missing.
7. In Data collection and processing, they stated that REFMAC4 was used. Is this correct? The current version is REFMAC5.

Detailed reply to the Reviewers' comments:

Reviewer #1 (Remarks to the Author):

Lentiviral Nef proteins interact with and modulate various cellular receptors, to facilitate viral replication and interfere with immune responses. Some cellular receptors, such as CD4, are targeted by both SIV and HIV Nef proteins, whereas other, such as CD3, are modulated by SIV and HIV-2 Nef but not by HIV-1. The structural bases for these differences are poorly characterized. Nef proteins are known to form oligomers. Manrique and colleagues solved the structure of SIV Nef bound to the ExxxLM motif of another Nef protein. They also reconstituted how Nef-CD4-AP2 complexes may be formed. Furthermore, by comparing SIV and HIV-1 Nef structures, they determined the structural elements necessary for the down-modulation of CD3. This structural analysis was combined to a functional study of a panel of HIV-1 Nef mutants. The activity of the mutants on various cellular receptors was analyzed when Nef was expressed alone, or within a full virus during infection of PBMC. This led to the characterization of key residues required for the activity of lentiviral Nef. I am unable to assess the structural analysis part because this is not my area of expertise. However, experiments are well performed and convincingly presented. It is clearly shown how the introduction of mutations in HIV-1 Nef rescues the CD3 down-modulation of this protein. This work provides new structural insights into how Nef proteins interact with different cellular receptors. I don't have major concerns.

We are pleased that this reviewer raised only minor issues and felt that our “*experiments are well performed and convincingly presented*”.

Minor comments

1. There are some minor discrepancies in the results obtained when Nef mutants are expressed as a single protein or from a proviral construct. For instance, Nef mutants D and S down-modulate CD3 when expressed alone, and not in infected cells. What are the overall amounts or stability of the mutants in the two experimental systems?

The effects of Nef-D and -S on CD3 surface expression in HEK293T cells were modest and statistically not significant ($p=0.20$ and $p=0.26$, respectively; one sample t test). Thus, they do not contradict our results obtained in infected primary cells. To further address this point, we performed Western blot analyses of both transfected HEK293T cells and infected PBMCs (new Supplementary Fig. 6a,b). Nef was detected using a rabbit antiserum raised against HIV-1 Nef that cross-reacts with SIVmac Nef. As now shown in Supplementary Fig. 6c, Nef expression levels in both experimental settings show a highly significant correlation ($R^2=0.87$, 0.0002).

2. HIV-1 Nef does not down-regulate CD3, but impairs the formation of the immune synapse, indicating that HIV-1 may have evolved other means to interfere with T cell activation. This could be discussed.

The reviewer raises an important but complex issue. Using primary human cells, we have previously shown that HIV-2 and SIV Nefs that down-modulate TCR-CD3 but not HIV-1 disrupt formation of the immune synapse between infected T cells and antigen-presenting macrophages or DCs (Arhel et al., JCI, 2009). However, another study reported that HIV-1 Nef causes a 2- to 3-fold reduction of conjugate formation between HIV-1-infected T lymphocytes and sAg-pulsed Raji B cells, resulting in reduced TCR signaling (Thoulouze et al., Immunity, 2006). In part, this discrepancy may result from

the use of primary DCs and macrophages versus Raji B cells. Notably, both studies found that HIV-1 Nef increases Lck accumulation in intracellular compartments. Whether HIV-1 Nefs reduce or increase the responsiveness of virally infected T cells to stimulation is the subject of a long-standing debate. However, as pointed out by this reviewer, HIV-1 Nefs also clearly deregulate the communication between virally infected T cells and APCs by modulating actin dynamics and downstream TCR signaling pathways. It is also well accepted, however, that SIV and HIV-2 Nef proteins that down-modulate CD3 disrupt immune synapse function and TCR signaling much more severely than HIV-1 Nefs. As suggested, this is now discussed in the revised manuscript (pg. 3, first paragraph).

3. The authors could discuss further the structural differences between SIV and HIV-2 Nef structures.

As now mentioned in the revised manuscript (pg. 6, first paragraph), SIVmac239 and HIV-2 Nef proteins are structurally highly similar because both viruses belong to the same lineage of primate lentiviruses and share a common origin, *i.e.*, SIVsmm infecting sooty mangabeys (reviewed in Sharp and Hahn, 2011). HIV-2 is the result of at least nine independent zoonotic transmissions and some HIV-2 strains are phylogenetically more closely related to SIVmac than to other HIV-2 isolates. Thus, there are no general structural differences between SIVmac and HIV-2 Nefs proteins, and our findings on SIVmac Nef are for most part presumably directly applicable to HIV-2 Nefs. We now clarify this point when describing the structure of SIV Nef:

“Notably, the Nef protein of SIVmac is structurally highly similar to HIV-2 Nefs because both viruses belong to the same lineage of primate lentiviruses and share a common origin (reviewed in Sharp and Hahn, 2011). HIV-2 is the result of at least nine independent zoonotic transmissions of SIVs infecting sooty mangabeys and some HIV-2 strains are genetically more closely related to SIVmac than to other HIV-2 isolates.”

Reviewer #2:

This work reports new X-ray crystal structures of an SIV mac239 Nef core protein alone and in complex with a modified human Hck SH3 domain. While the authors included the human CD4 cytoplasmic tail peptide, containing the ExxxLL endocytic motif required for Nef binding, no electron density was found for the CD4 peptide in these structures.

SIV Nef packed in the crystal with the C-terminal flexible loop helix of one Nef monomer making contacts with alpha helices 4 and 5 of an adjacent Nef in the trimer. Since this SIV Nef loop (and by extension the analogous HIV-1 Nef loop) houses an endocytic di-leucine motif, this interaction is used as a model for the endocytic ExxxLL motif recognition site for CD4, for which there is currently no structure for HIV-1 Nef. The hydrophobic pocket site on SIV Nef where the loop interacts is also the site where the CD3 zeta chain peptide interacts with a peptide dileucine motif as shown in a previous crystal structure of the SIV Nef mac239 core in complex with a CD3 tail peptide.

No HIV-1 Nef structure in complex with the CD4 tail peptide has been determined, and insight into that structure is important. However, the novelty of these SIV Nef X-ray structures is reduced by not having the CD4 peptide bound. The model based on the Nef:Nef interaction offers some insight, but doesn't really extend recent work by the Hurley group in terms of what the Nef:CD4:AP2 complex may look like at the plasma membrane (Ren et. al eLife 2014; citation below). Also HIV-1 Nef does not normally induce downregulation of CD3 zeta – thus the value of reconstituting this function by

extensive mutagenesis as presented in the paper does not provide new insight regarding HIV-1 Nef function.

We respectfully disagree with the notion that reconstitution of the CD3 down-modulation function “*does not provide new insight regarding HIV-1 Nef function*”. As previously published (e.g., Schindler et al., Cell 2006) and mentioned in the present manuscript, the loss of this Nef function distinguishes HIV-1 from most other primate lentiviruses and may contribute to its high virulence. Thus, understanding how this activity was lost and how it can be restored is clearly relevant for our understanding of HIV-1 Nef function.

With respect to the ternary CD4/Nef/AP2 complex formation we like to emphasize that in the Ren *et al.* paper the crystal structure of Nef with the $\alpha/\sigma 2$ hemicomplex of AP2 is reported. The study contains no insights into Nef-binding to an ExxxL ϕ endocytic sorting motif. The paper only describes the hydrophobic crevice of Nef as the potential binding site for the di-leucine based sorting motif and relates on NMR chemical shift perturbation data from the Grzesiek/Bax group in 1996 when describing where CD4 could fit into the Nef/AP2 complex (see Ren *et al.*, eLife 2014, Fig. 7). Importantly, no coordinate information is provided for this interaction as a structural basis was not known. We here determine in our study the first structure of a Nef protein binding to an ExxxLM motif, which we think very much extends the work from the Hurley group. This holds particularly well since binding of Nef to the $\alpha/\sigma 2$ hemicomplex only confirms previous structural data from the Owen group showing how the entire AP2 trunk complex interacts with a di-leucine based peptide (Kelly *et al.*, Nature, 2008). We therefore think that our study very well extends previous work on the Nef/AP2 complex formation by providing the structural basis for Nef binding to a dileucine-based sorting motif.

Specific Points:

1. Structural alignment and helical wheel analysis of Nef, CD3 and CD4 di-leucine motifs with Nef hydrophobic pockets suggests important residue contact information. Mutagenesis of new contact residues in the SIV and HIV-1 Nef hydrophobic pockets revealed in these structures should be performed to validate CD4 and CD3 di-leucine peptide binding, and extended to functional analyses in terms of receptor downregulation.

To address this point, we first took advantage of the HIV mutation database (<http://hivmut.org/>), which allowed us to identify previously published mutations within and around the hydrophobic pocket of HIV-1 Nef. In agreement with our findings, mutations of L114 (Gondim *et al.*, 2015), R109, R110 (Fackler *et al.*, 2006; Sauter *et al.*, 2015) as well as the combined mutation of I113, L116, Y119 and F125 (Poe and Smithgall, 2009) strongly reduced the ability of HIV-1 Nef to decrease CD4 surface levels. Importantly, Gondim and colleagues demonstrated that the single point mutation L114A (called L110A in their publication) is sufficient to abrogate the interaction of HIV-1 Nef with CD4. We now briefly discuss these functional analyses and highlight that they are in agreement with our structural models (pg. 13):

“In agreement with our model, not only mutations of R109 and R110, but also those of hydrophobic residues within the binding pocket have previously been shown to abrogate the ability of HIV-1 Nef to decrease CD4 surface levels. Intriguingly, FRET experiments by Gondim and colleagues demonstrated that mutation of Leu114, which interacts with the second leucine residue of the ExxxLL motif, fully disrupts the interaction of HIV-1 Nef with CD4.”

To further corroborate the role of the hydrophobic pocket and its basic rim in Nef-mediated down-modulation of CD4 and CD3, we also included functional analyses of Nef mutants from SIVmac, SIVsmm and HIV-2 in Supplementary Fig. 5 of our manuscript. In agreement with our model, flow cytometric analyses of Jurkat T cells revealed that the di-arginine motif is required for efficient Nef-mediated down-modulation of CD4 and CD3, but not MHC-I (pg. 13/14):

“Furthermore, we here expand these functional analyses to CD3 and analyzed mutations of R137 and R138 in SIVmac, SIVsmm and HIV-2 Nef. Our finding that the respective mutants lose their ability to downmodulate CD3 and CD4, but not MHC-I, supports an important role of the interaction of these two residues with the acidic residue in the ExxxLL motif.”

2. It is not clear why the authors cut off the X-ray data at 2.6 angstroms for the Nef core structure since mean I/sigma in the highest resolution shell is 3.6; the X-ray data are likely to be useful to higher resolution. It is common practice to cutoff x-ray data to an I/sigma of 2 or 1.5.

We thank the reviewer for this helpful comment. We previously took a conservative approach and cut the resolution to have R_{meas} at highest resolution shell below 70%. Following the suggestion of reviewer #2, we cut off data at an I/sigma of 2.01 (though, R_{meas} at the highest resolution shell came close to 100%), which eventually not only improved the resolution to 2.50 Ang, but also reinforced the electron density map.

3. The difference in the Rwork/Rfree refinement values of 7% is unacceptable for structure refinement and may be an indication of model bias. The authors need to improve the structure refinement by including refinement using simulated annealed omit maps.

With improved data quality, we could now build the peptide chain more confidently. Reprocessing and cleaning data at 2.50 Ang also helped in lowering the difference between R_{work}/R_{free} (now ~5%). Looking at the difference-electron density map for the peptide and improvement in Ramachandran plot, we accept 5% difference between R_{work}/R_{free} as useful guide. We have built 49 water molecules and almost all side-chains which have sigma level below one were removed from the refinement by keeping their occupancies zero.

4. The number of residues with phi/psi angles in the allowed region of the Ramachandran plot is much too large. This is another indication that the structure needs to be better refined.

Upon refinement, following the suggestion of reviewer #2, the percentage of residues in the most favored region now improved to 96.28% and the portion of residues in the allowed region is the remaining 3.72%. There are no 'Ramachandran outliers' in the structures.

5. A few aspects of the flow cytometry data require further clarification (Figure 5A and 6). What are the 'control' plots – presumably these cells are uninfected and not stained with antibodies. This should be explicitly indicated.

The control plots show cells that were transfected (Fig. 5A) or infected (Fig. 6) with HIV-1 *nef* stop IRES eGFP. They were stained with isotype control antibodies to determine unspecific antibody binding and background fluorescence. To clarify this point, we have labeled the control plots with “*nef* stop, isotype control” in the revised manuscript and now explain in the methods section (pg. 19) that the background fluorescence obtained with the isotype control antibodies was subtracted from all stained samples:

“Flow cytometric analysis

To determine the effect of Nef on CD4 and CD3, HEK293T cells were transfected by the calcium phosphate method with 5 µg of an expression vector for *nef* or the respective vector control expressing only eGFP and either 1 µg of an expression plasmid for CD4 or 0.2 µg of an expression plasmid for CD3ζ-CD8. Two days post transfection, cells were stained for CD4 (MHCD0405, Life Technologies) or CD8 (555369, BD) and analyzed by two-color flow cytometry. Cell surface receptor downmodulation was calculated as described previously⁶⁰. Briefly, the mean fluorescence intensities were determined for cells showing specific ranges of GFP expression. Isotype control antibodies were used to determine background fluorescence and unspecific antibody binding. The mean fluorescence intensities obtained with the isotype control antibodies were subtracted from those obtained with specific surface receptor antibodies. The fluorescence values obtained for cells transfected with the control construct expressing only GFP was compared with the corresponding value obtained for cells co-expressing Nef and GFP. The same regions of GFP expression were used in all calculations.”

In addition, a second (or alternative) control of uninfected cells stained with both antibodies should be included for comparison.

A double staining of uninfected cells is not meaningful since cells were only stained for the respective surface receptors (y axis). As outlined in the figure legend, the green fluorescence signal (x axis) is the result of an eGFP reporter gene that is directly expressed from the HIV-1 constructs used. To better illustrate our eGFP gating strategy, we included an untransfected control in the revised version of Fig. 5a, that was stained for CD3 or CD4.

The gating strategy is not completely clear especially along the y (receptor) axis. In Figure 5A, none of the panels seem to show a significant signal with the CD3 antibody.

We respectfully disagree with the reviewer in this point. As indicated in each plot, the mean fluorescence intensities obtained with the anti-CD3 antibody vary between 1101 and 9745, and are thus about 10- to 100-fold higher than the background signal obtained with the respective isotype control antibody. We now explain in the legend of Fig. 5 that the numbers in the FACS plots represent mean fluorescence intensities of the respective surface receptor staining.

“Figure 5. Gain-of-function mutations in HIV-1 Nef for CD3 internalization.

HEK293T cells were cotransfected with expression vectors for CD3ζ-CD8 or CD4 and constructs co-expressing the indicated *nef* alleles and eGFP via an IRES and examined by flow cytometric analysis two days post transfection. **(a)** Examples for primary FACS data with the gating strategy and the mean fluorescence intensity (MFI) of the stained surface receptor and **(b)** overview on Nef proteins examined and their CD3 and CD4 downmodulation activities. Bars on the left illustrate the composition of the Nef chimeras A to T. Domains derived from HIV-1 SF2 and SIV_{mac239} Nef are highlighted in beige and blue, respectively. Numbers on top indicate the respective amino acid positions. Red arrows indicate the Nef proteins that were selected for further analyses in infected primary cells (Fig. 6). Values on the right are normalized to the *nef*-deficient vector control (100%) and represent means of two to four independent experiments (± SEM). Numbers in panel a provide mean fluorescence intensities (see also Supplementary Figure 6).”

Where is the cutoff to determine the number CD3-positive cells? The same issue is present for Figure 5a with CD4.

To quantify the effect of Nef on receptor surface expression, we only gated for eGFP positive cells, *i.e.*, cells expressing HIV-1. The respective gate is indicated in all FACS plots shown. We did not determine the percentage of receptor positive cells but rather determined the mean fluorescence intensities (MFI) of the respective receptor staining. Calculating the MFI is less biased and may also detect weak effects of Nef, while the percentage of receptor expressing cells does not allow

distinguishing between weakly and strongly positive cells. Thus, the background fluorescence determined via isotype control staining was subtracted, but no cut-offs on the y axes were set.

Also the number of CD4-positive cells seems to increase in the presence of increasing amounts of HIV-1 SF2 Nef, whereas the opposite would make more sense. Having the uninfected/double-stained controls should help here. Cut-offs for each flow panel in Figure 6 should also be added; contour plots may help as well.

The increase of CD4 positive cells with increasing amounts of HIV-1 SF2 Nef is due to the experimental setup: As HEK293T cells do not endogenously express CD4, they were co-transfected with the indicated HIV-1 clones and an expression plasmid for CD4. Thus, cells that are efficiently transfected do not only express more Nef, but also more CD4. In agreement with this, cells expressing more HIV-1 nef *stop* also expressed more CD4. Importantly, this effect was not observed in infected primary cells (Fig. 6c) and transfected Jurkat cells (Supplementary Fig. 5) as they endogenously express CD4. As outlined above, we did not set a cut-off for CD4, as we feel that the MFI of CD4-APC within the HIV-1-positive population is more informative than the percentage of CD4 positive cells.

Reviewer #3:

The Nef protein of lentivirus facilitates internalization of CD3, CD4 or other immune receptors by recruitment of AP complex to induce clathrin-dependent endocytosis. Internalization of CD3 or CD4 in regular situations (i.e., without the Nef protein) is coupled with phosphorylation of their intracellular regions, whereas their Nef-mediated internalizations do not require such phosphorylation. The Nef proteins can recognize non-phosphorylated internalization motifs and also interact with the AP complex through their own dileucine-based sorting motifs. The Nef proteins can act as adaptors linking the AP complex and immune receptors. The internalization of CD4 mediated by the Nef proteins of HIVs and SIVs may promote the release of fully infectious virus particles and prevent super-infection. On the other hand, CD3 can be internalized by the Nef proteins of HIV-2 and most SIVs but not by that of HIV-1. The recognition manner of the tyrosine-based sorting motif of CD3 by SIV Nef has been reported, whereas that of the dileucine-based motif of CD4 has remained unclear. In this study, Manrique et al. aimed to reveal mechanisms for recognition of the dileucine-based sorting motif of CD4 by Nefs of HIV-1 and SIV. However, they failed to crystallize the complex between CD4 and HIV-1 or SIV Nef. Instead, they obtained the SIV Nef structure, where the dileucine-based sorting motif of one SIV Nef (for the interaction with the AP2 complex) is recognized by another adjacent SIV Nef in the crystal. The dileucine-based motif of SIV Nef (ED'E'EHY'LM'HP) may mimic that of CD4 (MS'Q'ILR'LL'SE). This idea aided their subsequent mutational analyses to investigate how the tyrosine-based motif of CD3 and the dileucine-based motif of CD4 are discriminated, together with the previously reported complex structure between SIV Nef and CD3ζ.

The crystallography and binding experiments using ITC appears to be properly done. The cell-based internalization assays may be consistent with the structure and results of the binding experiments. However, hydrophobic-interaction-based binding might be promiscuous and interacting residue pairs can be changed in some cases. I am not confident that the recognition of 'EEHYLM' from SIV Nef is (basically) identical to that of 'QILRLL' from CD4. In addition, the N-terminal YxxL motif of SIV Nef makes the interpretation of the results from the internalization assay complicated. Overall, the key mechanism for discrimination between the tyrosine- and dileucine-based sorting motifs by Nef is not clear. The findings in this study seem to be of interest in the limited field.

We thank the reviewer for his/her evaluation of the study but like to respectfully disagree about the interest in the field. Since “Serine phosphorylation-independent downregulation of cell-surface CD4 by Nef” has been reported by Garcia and Miller in 1991 (*Nature* **350**, 508-511; same title) and the finding that “Nef induces CD4 endocytosis: Requirement for a critical dileucine motif in the membrane-proximal CD4 cytoplasmic domain” by Aiken *et al.* (*Cell* **76**, 853-864, 1994), the structural basis for this interaction is missing. This is most likely due to a very low affinity between these two molecules, as it is often observed for protein–protein interactions at–or close to–the plasma membrane. Of note, the dissociation constant between Nef and a CD4 peptide was determined to 500 μ M (Grzesiek *et al.*, *Biochemistry* 1996). Nef binding to T cell receptors became again a very important topic in HIV research when differences between CD3 and CD4 internalization by the Nef protein could be correlated with prospects of pathogenic or non-pathogenic viral infections (Schindler *et al.*, *Cell* **125**, 2006). The enigmatic observation here is that some Nefs are able to bind both tyrosine and dileucine based sorting motifs of T cell receptors (the non-pathogenic SIV and HIV-2 strains), whereas pathogenic HIV-1 strains lost the ability of CD3 internalization and only interact with dileucine-based sorting motifs. Regarding the question if the ExxxLM motif in SIV Nef fully mimics binding of the SQxxxLL motif in CD4 to Nef we can at least confirm that our structural data are in complete agreement with the NMR chemical shift data reported by Grzesiek *et al.* on the interaction between HIV-1 Nef and a 13-residue long peptide (MS-QIKRLLSEKKT) that contains the CD4 di-leucine based internalization motif. Particularly residues G95, G96, L97, R106, and L110 of HIV-1 Nef_{NL4-3} reported in that study, which correspond to residues G127, G128, L129, R138, and L142 in SIV_{mac239} Nef, are also identified as the interaction surface for ExxxL ϕ motifs in our study (see Figs 2 and 3). We are well aware that other sites and motifs, as the N-terminal YxxL motif in SIV Nef, might additionally contribute to the affinity and specificity of the Nef – T cell receptor tail interaction at the plasma membrane. However, in the absence of any other structural data of this important virus – host cell protein interaction, our study provides the very first structural insights into Nef binding to an ExxxLM motif.

Finally, I would like to suggest several points to be improved as follows:

1. Hexagonal or trigonal assembly of Nef in the crystal may be artifact in crystallization, if there is no finding suggesting that similar assemblies occur in cells or at least in solution. Fig. 1a and b should be removed or moved to Supplementary information. In Fig. 1d, a type of electron density map should be described (probably 2Fo-Fc map).

The figure displaying the hexagonal assembly of Nef in the asymmetric unit has now been removed. In addition, a stereo image displaying the 2F_o-F_c electron density map of the ExxxLM motif binding to the Nef hydrophobic crevice has been included in the revised Fig 1c to address the reviewers’ concern and to comply with the editorial polices.

2. In Fig. 2, I prefer stick and cartoon models for presentation of protein-protein interactions. Electrostatic surface is not so informative. Fig. 2c could be moved to Supplementary information.

We respectfully like to disagree with the reviewer on the necessity to show the electrostatic surface of the sorting motif recognition site in Nef. As this interaction is taking place at the membrane, we think it is very important to show that the binding site is surrounded by positively charged residues. These residues may orient the myristoylated Nef protein to the negatively charge plasma membrane and make the binding site accessible to the receptor tail sequences. The assembly of the endocytosis active complex and the conservation of basic residues is discussed in the second paragraph of the

Discussion section.

3. In Fig. 4b, mutated residues are not easily recognized. The SIV and HIV residues may be presented in different colors.

We agree with the reviewer and changed the colors of the mutant accordingly. They are now displayed in Fig. 4b in an inverse white/blue manner and can be easily recognized.

4. In the left panel of Fig. 5, explanation of mutants 'A–T', color bars and red arrows should be added in the figure legend.

We changed the figure legend as suggested:

“Figure 5 Gain-of-function mutations in HIV-1 Nef for CD3 internalization.

HEK293T cells were cotransfected with expression vectors for CD3z-CD8 or CD4 and constructs co-expressing the indicated nef alleles and eGFP via an IRES and examined by flow cytometric analysis two days post transfection. (a) Examples for primary FACS data with the gating strategy and the mean fluorescence intensity (MFI) of the stained surface receptor and (b) overview on Nef proteins examined and their CD3 and CD4 downmodulation activities. Bars on the left illustrate the composition of the Nef chimeras A to T. Domains derived from HIV-1 SF2 and SIV_{mac239} Nef are highlighted in beige and blue, respectively. Numbers on top indicate the respective amino acid positions. Red arrows indicate the Nef proteins that were selected for further analyses in infected primary cells (Fig. 6). Values on the right are normalized to the *nef*-deficient vector control (100%) and represent means of two to four independent experiments (\pm SEM). Numbers in panel a provide mean fluorescence intensities (see also Supplementary Figure 6).”

5. The letters representing cell constants (*a*, *b*, *c*, *alpha*, *beta*, *gamma*) and the initial letter of the space group should be italicized.

This has been corrected.

6. PDB codes are missing.

We submitted the two crystal structures of SIV_{mac239} Nef and the SIV_{mac239} Nef–Hck_{SH3} complex to the protein data bank with accession numbers 5NUI and 5NUH, respectively. The PDB codes are now listed in the footnote (pg. 20) and the crystallographic table (Supplementary Table 1):

“Data availability. Structure factors and coordinates for SIV_{mac239} Nef and the SIV_{mac239} Nef–Hck_{SH3} complex have been deposited in the Protein Data Bank with accession codes 5NUI and 5NUH, respectively. All other data supporting the findings of this study are available from the corresponding authors upon request.”

7. In Data collection and processing, they stated that REFMAC4 was used. Is this correct? The current version is REFMAC5.

We thank the reviewer for the attentive comment. Indeed, the current version REFMAC5 has been used for data collection and processing, which is now corrected in the manuscript (pg. 17):

“Alternating cycles of manual building in COOT⁵⁶ and refinement in REFMAC5⁵⁸ and Phenix⁵⁵ were performed to generate the final structural model.”

Following the comments from another reviewer (see reviewer 2, point 2), we re-processed our data to improve the quality and resolution. For this re-processing of the data we used Phenix only.

REVIEWERS' COMMENTS:

Reviewer #1 (Remarks to the Author):

the authors have addressed my previous concerns

Reviewer #2 (Remarks to the Author):

The authors have done a credible job responding to the previous comments. In particular, they have further refined and improved the quality of their structures and the presentation of the structural models in the revised manuscript.